# Determinants of mosaic chromosomal alteration fitness

Yash Pershad [1,33], Taralynn Mack[1,33], Hannah Poisner [1], Yasminka A. Jakubek[2], Adrienne M. Stilp [3], Braxton D. Mitchell [4], Joshua P. Lewis [4], Eric Boerwinkle[5], Ruth J. F. Loos [6], Nathalie Chami [6], Zhe Wang [6], Kathleen Barnes [7], Nathan Pankratz [8], Myriam Fornage [5,9], Susan Redline[10], Bruce M. Psaty [11], Joshua C. Bis [12], Ali Shojaie[13], Edwin K. Silverman[14], Michael H. Cho[14], Jeong H. Yun [14], Dawn DeMeo[14], Daniel Levy[15], Andrew D. Johnson [15], Rasika A. Mathias [16], Margaret A. Taub[17], Donna Arnett[18], Kari E. North [19], Laura M. Raffield [20], April P. Carson [21], Margaret F. Doyle[22], Stephen S. Rich [23], Jerome I. Rotter [24], Xiuqing Guo [24], Nancy J. Cox[25], Dan M. Roden [26], Nora Franceschini [19], Pinkal Desai[27], Alex P. Reiner [28], Paul L. Auer [29], Paul A. Scheet [30], Siddhartha Jaiswal [31], Joshua S. Weinstock [32] & Alexander G. Bick [25] ✉

Clonal hematopoiesis (CH) is characterized by the acquisition of a somatic mutation in a hematopoietic stem cell that results in a clonal expansion. These driver mutations can be single nucleotide variants in cancer driver genes or larger structural rearrangements called mosaic chromosomal alterations (mCAs). The factors that influence the variations in mCA fitness and ultimately result in different clonal expansion rates are not well understood. We used the Passenger-Approximated Clonal Expansion Rate (PACER) method to estimate clonal expansion rate as PACER scores for 6,381 individuals in the NHLBI TOPMed cohort with gain, loss, and copy-neutral loss of heterozygosity mCAs. Our mCA fitness estimates, derived by aggregating per-individual PACER scores, were correlated ($R^2 = 0.49$) with an alternative approach that estimated fitness of mCAs in the UK Biobank using population-level distributions of clonal fraction. Among individuals with *JAK2* V617F clonal hematopoiesis of indeterminate potential or mCAs affecting the *JAK2* gene on chromosome 9, PACER score was strongly correlated with erythrocyte count. In a cross-sectional analysis, genome-wide association study of estimates of mCA expansion rate identified a *TCL1A* locus variant associated with mCA clonal expansion rate, with suggestive variants in *NRIP1* and *TERT*.

With age, hematopoietic stem cells (HSCs) accumulate somatic mutations[1]. While most of these mutations have little effect on cell fitness and become simply passengers, some mutations, called drivers, increase the fitness of a HSC and lead to clonal expansion. The phenomenon of clonal hematopoiesis (CH) occurs when a clone of cells is detectable without causing cytopenias, dysplastic hematopoiesis, or hematologic malignancy[2]. CH is common in the elderly and prevalence increases with age. Previous studies have shown that CH is associated with increased risk of all-cause mortality and many common complex diseases[3–11].

Mutations in HSCs causing CH include single nucleotide variants (SNVs) in genes associated with hematological malignancies (e.g., *DNMT3A, TET2,* and *JAK2*)[12] referred to as Clonal Hematopoiesis of Indeterminate Potential (CHIP) and larger chromosomal rearrangements called mosaic chromosomal alterations (mCAs)[9]. These mutations do not include non-detectable translocations or alterations to methylation profiles that can also result in clonality. mCAs, which may involve a gain or loss of a > 1 Mb segment of a chromosome or a copy-neutral loss of heterozygosity (CN-LOH), occur in between 10-20% of individuals over 55 years old without cancer[4,13]. Specific sets of mCAs have been associated with risk of lineage-specific hematologic malignancies[14,15]. Although individuals with CH clones of higher clonal fraction (i.e., larger clone size) generally have worse health outcomes, the factors that influence the variations in mCA fitness and ultimately result in different clonal expansion rates have not been studied and are not well-understood[5,10,16,17].

Conventional methods to study clonal fitness require collecting serial blood samples over several decades, so obtaining sufficient sample sizes from existing biobanks is challenging as they typically only include one blood sample per individual. To overcome this limitation, methods have been developed to estimate the expansion rate of a clone in an individual from a single blood draw. Watson and Blundell estimated fitness of SNVs and mCAs on a population level from clonal fraction distributions in UK Biobank participants[18,19]. However, this method only predicts clonal expansion rate for a mutation aggregated over a population rather than for a mutation in a specific individual. Estimating the clonal expansion rate of a given driver mutation in an individual is essential to elucidate germline modifiers of clonal expansion and better characterize the pathophysiology of CH-associated diseases.

We recently developed a method called passenger-approximated clonal expansion rate (PACER), which uses the abundance of passenger mutations accompanying a driver mutation to estimate the clonal expansion rate of a SNV with a single blood sample from an individual[20]. Here, we apply PACER to 6,381 individuals with gain, loss, and CN-LOH mCAs, detected by whole genome sequencing (WGS), in the NHLBI Trans-Omics for Precision Medicine (TOPMed) dataset to calculate a PACER score per individual and identify determinants and consequences of mCA clonal expansion rate (Fig. 1A). PACER estimates of mCA fitness, derived by calculating the median of PACER scores across all individuals with a specific mCA, were compared to the fitness of SNV mutations implicated in CHIP (Supplementary Fig. 1). We examined associations between PACER score and peripheral blood counts and observed that for individuals with single nucleotide variants or mCAs affecting *JAK2*, higher PACER score (i.e., faster mCA expansion rate) associated with higher erythrocyte counts. Next, we performed a genome-wide association study (GWAS) of PACER score among individuals with a single mCA and found that variants in *TCL1A*, *NRIP1*, and *TERT* may modulate mCA clonal expansion.

## Results

We identified 6930 people with one mCA and 763 with multiple mCAs. After excluding mCAs of sex chromosomes, we detected 3828 autosomal mCAs in 3068 unique individuals (Fig. 1B). 571 individuals with detected autosomal mCAs also had detectable sex chromosomal mCAs, where 369 were loss of chromosome X and 202 were loss of chromosome Y. The median age of individuals with autosomal mCAs was 67, and 57% were female. The most common autosomal mCAs detected were on chromosomes 11, 1, and 9. Of detectable mCAs, CN-LOH mCAs were the majority (52%). mCAs within the same type and chromosome exhibited high variability in clonal fraction (Fig. 1C). Visualization of the mCAs detected in TOPMed is shown in Fig. 1 of Jakubek et al.[21].

### Passenger-approximated clonal expansion rate of mCAs by chromosomal event

We calculated passenger mutation counts, representing clock-like C > T or T > C somatic mutations (see Methods), in individuals with a single mCA and no CH-associated SNVs[20]. Within this cohort, the minimum total passenger mutation count for a patient with an mCA was 3, maximum was 933, and median was 53 (Fig. 2A). We then calculated a covariate-adjusted PACER score to approximate the clonal expansion rate for the mCA of each individual from the normalized residuals of a negative binomial regression of age, sex, and clonal fraction predicting total passenger mutation count.

Compared to SNVs, mCAs involve changes to larger regions of chromosomes. To check for potential confounders, we investigated if these larger rearrangements alter total passenger mutation counts, thereby confounding our estimations of clonal expansion rate. To test this, we compared passenger mutation counts between the chromosome containing the mCA to the chromosomes not containing the mCA and found no significant difference by mCA location and type (Supplementary Fig. 2A) or in aggregate ($p = 0.12$). Furthermore, we found that PACER scores did not significantly change upon exclusion of the mCA chromosome. A linear regression of the covariate-adjusted PACER scores with and without the mCA chromosome demonstrated a high degree of correlation, with a Spearman's rank correlation coefficient ($\rho$) of 0.82 (Supplementary Fig. 2B). Moreover, a linear regression of covariate-adjusted PACER scores with and without a random chromosome had the same correlation, with a $\rho$ of 0.82 (Supplementary Fig. 2C).

We computed the median PACER scores for individuals with the same mCA type and location to derive a measure of an mCA's fitness from PACER. With this estimation of mCA fitness derived from PACER scores, we then calculated the fold-change relative to a loss of chromosome X, defined as a loss of a > 100 Mb segment of chromosome X (Fig. 2B). Gain of chromosome 1, gain of chromosome 7, loss of chromosome 14, and CN-LOH of chromosome 14 had the highest PACER-derived mCA fitness. However, of those mCAs, only CN-LOH of chromosome 14 occurred in more than 25 individuals in TOPMed. Of all mCAs, only a gain in chromosome 1 had a higher fitness than driver SNVs in non-R882 *DNMT3A*, which is the slowest growing CHIP mutation[20].

To validate our estimates, we compared our PACER-estimated fitness by mCA in TOPMed to those generated by another approach based upon clonal fraction distribution in the UK Biobank[18]. A generalized linear model of mCA fitness derived from PACER scores and median age of individuals in TOPMed explained 49% of the variance in this clonal-fraction-derived (CF-derived) fitness in the UK Biobank (Fig. 2C), compared to 14% for a model using only median age and 44% for a model using only median total passenger mutation counts. mCA fitness derived from PACER scores had a significant positive association with CF-derived fitness ($\beta = 0.04$, 95% CI = [0.017, 0.063], $p = 0.001$).

We next investigated the fitness of two curated sets of mCAs associated with future development of either myeloid or lymphoid hematologic malignancy[14]. In TOPMed, the lymphoid set of mCAs had the highest covariate-adjusted PACER scores, followed by the myeloid mCAs and then mCAs associated with neither malignancy (Supplementary Fig. 3). ANOVA demonstrated a significant difference in PACER-estimated fitness among these three groups ($p = 0.0048$).

### PACER scores associate with erythrocyte counts

To understand the relationship between clonal expansion rate with clinical phenotypes, we also investigated the association between PACER score and peripheral blood counts in individuals with mCAs. Blood counts were available for 987 individuals with mCAs.

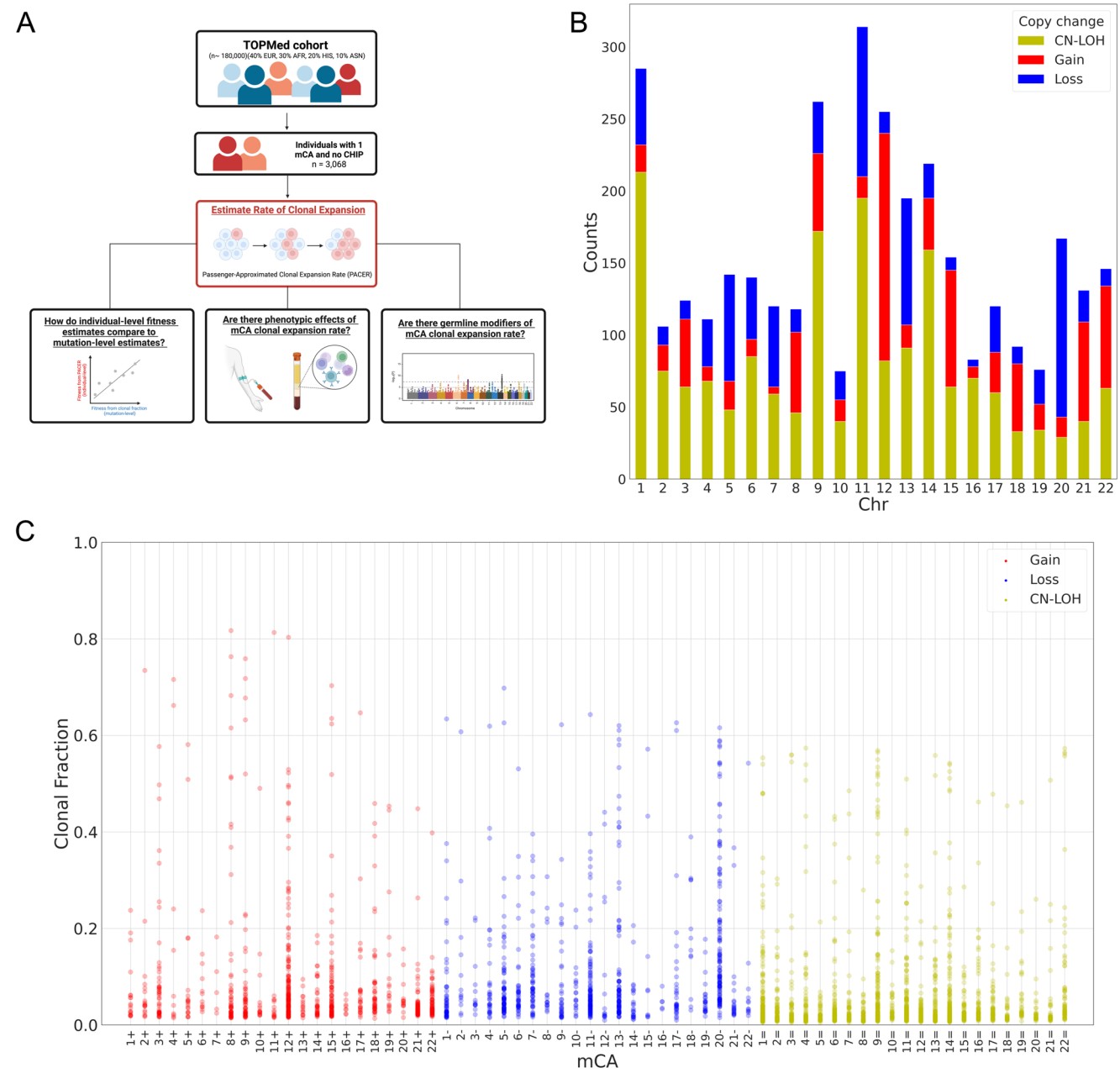

**Fig. 1 | Schematic of the study and mosaic chromosomal alterations in TOPMed. A** Excluding mosaic chromosomal alterations (mCAs) of chromosome X, 3068 individuals had 3,828 mCAs in TOPMed. 6,930 individuals had 1 mCA, and 763 had > 1 mCA. Created with BioRender.com released under a Creative Commons Attribution-NonCommercial-NoDerivs 4.0 International license. **B** Stacked bar plot showing counts of mCAs by chromosome, separated by copy change type: copy neutral loss of heterozygosity (CN-LOH) in yellow, gain in red, and loss in blue. mCAs of chromosome X were excluded. **C** Dot plot of clonal fractions for each patient with specific mCA (chromosome and copy change). Red and + represents gain of chromosome, blue and - represents loss of chromosome, and yellow and = represents CN-LOH.

Among individuals with lymphoid mCAs, a multiple regression of age at time of blood draw, sex, clonal fraction, and PACER score predicting lymphocyte count explained 13.3% of the variance in lymphocyte counts, compared to only 2.5% for a model without PACER score (Fig. 3A). In the multiple regression, PACER score had a significant positive association with lymphocyte count (β = 0.0175, 95% CI [0.003, 0.032], p = 0.019). In contrast, for individuals with myeloid mCAs, the same regression explained 3.5% of the variance in myeloid cell counts; PACER score was not associated with myeloid cell count (β = −0.0031, 95% CI [−0.009, 0.003], p = 0.298) (Fig. 3B).

For the 23 individuals with lymphoid mCAs who had lymphocyte counts, a multiple regression of age at time of blood draw, sex, clonal fraction, and PACER score predicting lymphocyte count, demonstrated a significant association between PACER score and lymphocyte counts (β = 0.0175, 95% CI [0.003, 0.032], p = 0.019). However, after excluding outliers with lymphocyte count greater than $10 \times 10^9$ cells/L, there was no significant association. These outliers may represent elevated lymphocyte counts in three individuals with a high PACER score due to rapidly expanding mCA clones.

For the 47 individuals with myeloid mCAs who had myeloid cell counts, a multiple regression of age at time of blood draw, sex, clonal fraction, and PACER score predicting myeloid cell count found that PACER score was not significantly associated with myeloid cell count (β = −0.0031, 95% CI [−0.009, 0.003], p = 0.298).

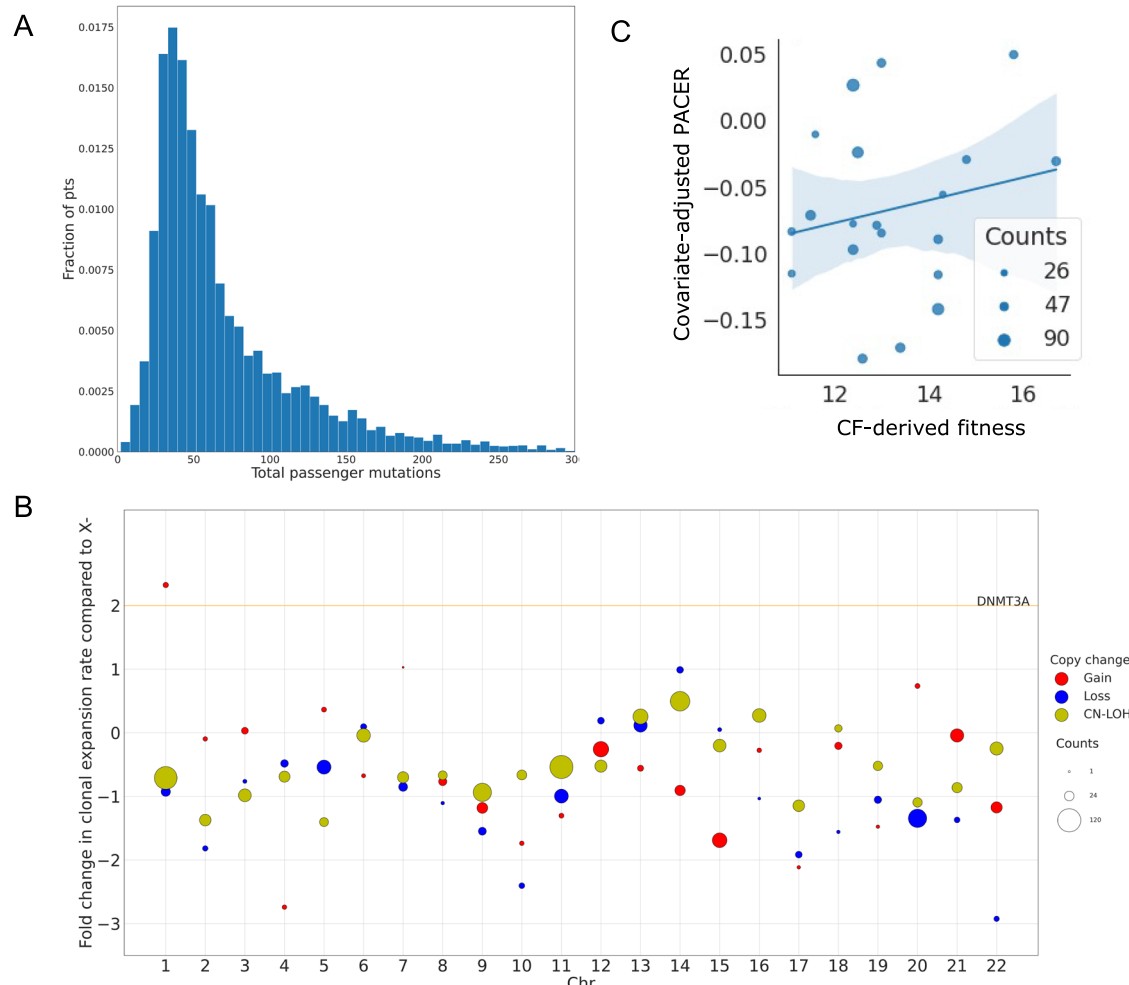

**Fig. 2 | Total passenger mutations and PACER score of mCAs. A** Histogram of total passenger mutations for all individuals with 1 mosaic chromosomal abnormality in TOPMed. A passenger mutation is defined as a clocklike C > T or T > C substitution that does not occur in a CHIP-associated gene. **B** Dot plot of fold change in clonal expansion rate compared to loss of chromosome X (X-). Red dots represent a gain of chromosome, blue dots represent a loss of chromosome, and yellow dots represent CN-LOH. The PACER scores are calculated after covariate adjustment (age, sex, study cohort, and clonal fraction) and inverse normalization of the total passenger mutations for all individuals with a single mCA. The median of the PACER scores is computed for individuals with the same mCA type and location to estimate mCA fitness. The fold change in estimated mCA fitness is calculated by dividing the clonal expansion rate for a given mCA by that for a loss of chromosome X. The size of the dot corresponds to the number of individuals with that mCA type. The orange line represents the fold change of clonal expansion rate of the CHIP mutation *DNMT3A* with respect to X. **C** Scatter plot of PACER-estimated mCA fitness and fitness derived from clonal fraction by mCA by Watson and Blundell 2023 (CF-derived fitness) for mCAs with >25 individuals with a given mCA type. The size of the dot corresponds to the number of individuals with that mCA type. A generalized linear model of PACER score, mean age, and mean clonal fraction predicting CF-derived fitness had an R$^2$ value of 0.49. The translucent bands around the linear regression line represents a bootstrap-estimated 95% confidence interval.

Since mutations in *JAK2*, such as *JAK2* V617F, on chromosome 9p are known to cause polycythemia vera, we performed a multiple regression of age at time of blood draw, sex, and PACER score to predict erythrocyte counts for 11 individuals with CN-LOH or loss of chromosome 9p and erythrocyte count. This model explained 91.6% of the variance in erythrocyte count, and PACER score had a significant positive association with erythrocyte count ($\beta = 0.0119$, 95% CI [0.006, 0.018], $p = 0.018$) (Fig. 3A).

We then examined the association between PACER score and erythrocyte counts among individuals with *JAK2* V617F clonal hematopoiesis of indeterminate potential (CHIP). We performed multiple linear regression of age at time of blood draw, sex, clonal fraction, and PACER to predict erythrocyte count among individuals with *JAK2* V617F CHIP and observed a significant association ($\beta = 0.341$, 95% CI [0.133, 0.537], $p = 0.001$), and the model with PACER score included had an R$^2$ of 0.90 while a model with only age and sex had an R$^2$ of 0.32. Therefore, for individuals with mCAs or

somatic SNVs affecting *JAK2*, erythrocyte count is associated strongly with PACER score.

**Germline genetic determinants of mCA clonal expansion rate**
The high variability in clonal expansion rate across a wide range of individuals with the same mCA demonstrates that other factors, including germline mutations and environmental exposures likely affect mCA clonal expansion rate. To identify germline variants associated with PACER score, we performed a GWAS of total passenger mutations in 6,381 individuals with 1 mCA (including mCAs of chromosome X) and without known CH driver SNVs (Fig. 4A). We controlled for age, sex, ancestry, study cohort, and clonal fraction. The GWAS identified a single nucleotide polymorphism (SNP) on chromosome 14, rs1122138, with genome-wide significance (p-value = 3.1 × 10-8). SNP rs1122138 is in an intronic region of *TCL1A*, and the alternate A allele is common, occurring in 21% of the haplotypes sequenced in TOPMed. For the leading variant in *TCL1A*, we observe a statistically

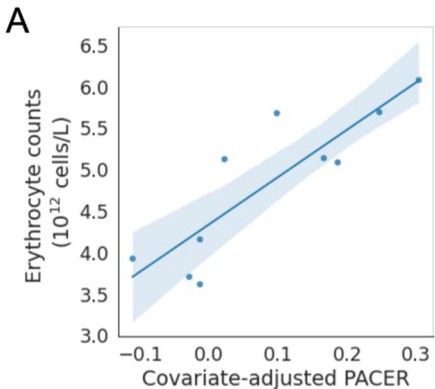
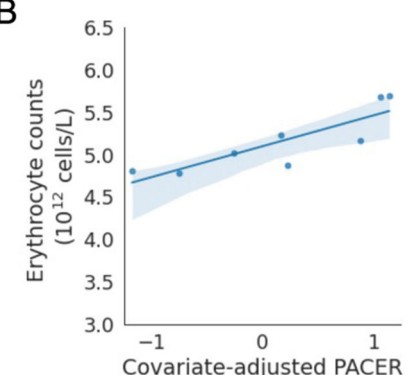

**Fig. 3 | Correlation of PACER score and peripheral erythrocyte counts.** Scatterplot of erythrocyte counts ($10^{12}$ cells/L) versus covariate-adjusted passenger associated clonal expansion rate (PACER) score for **A** patients with copy-neutral loss of heterozygosity or loss of the p arm of chromosome 9 and **B** patients with *JAK2* V617F clonal hematopoiesis of indeterminate potential. The translucent bands around the linear regression line represents a bootstrap-estimated 95% confidence interval.

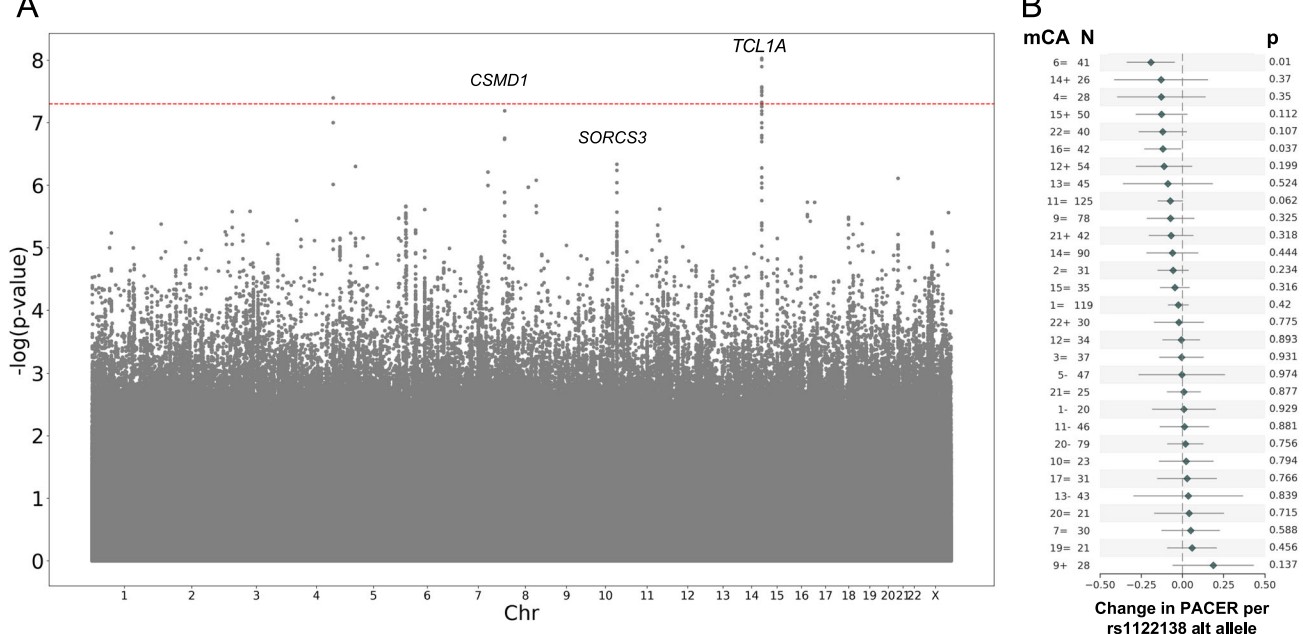

**Fig. 4 | Genome-wide association study of PACER score. A** Manhattan plot from the genome-wide association study (GWAS) for passenger-approximated clonal expansion rate (PACER) for single nucleotide polymorphisms (SNPs) with a minor allele frequency > 1%. A linear mixed model with kinship adjustment was used to regress inverse normally transformed total passenger mutations, with age, sex, clonal fraction, study, and the first 10 ancestral principal components included in the model as covariates. The dashed red line represents $5 \times 10^{-8}$, our Bonferroni multiple-hypothesis correction *p*-value threshold for significance. Nearest gene is labeled. SNPs within *TCL1A* had a p-value of $3.1 \times 10^{-8}$, those within *CSMD1* had a p-value of $1.4 \times 10^{-7}$, and those within *SORCS3* had a p-value of $4.6 \times 10^{-7}$. **B** Forest plot of change in PACER score per rs1122138 allele count and p-value for in a multiple regression model of age at blood draw, sex, and clonal fraction to predict PACER, with number of individuals with that mCA labeled as N. Data are presented as mean values ± 1.96*SE.

significant protective effect, with a negative effect estimate in a regression of PACER score, for sub-analyses excluding individuals with mosaic loss of chromosome Y and in only females.

The SNP rs1122138 and another SNP in the core promoter of *TCL1A*, rs2887399, which has been reported to modulate stem cell expansion for SNV CH[20], are in high linkage disequilibrium ($R^2 = 0.826$) in the 1,000 Genome Project. The risk alleles, rs1122138(A) and rs2887399(T), were correlated and our conditional analysis of our GWAS summary statistics demonstrated that rs1122138 did not remain significant after adjusting for the effect of rs2887399. This *TCL1A* locus was previously associated with the cross-sectional prevalence of mCAs, suggesting the underlying mechanism for this locus is related to clonal expansion[13,17].

To identify possible mCA-specific effects of rs1122138 on PACER score, we performed a multiple regression of age, sex, clonal fraction, and rs1122138 alternate allele count predicting total passenger mutation count for individuals with each mCA type. The coefficient for the rs1122138 alternate allele count varied by mCA type, but was not significantly associated with PACER score for any mCA type after multiple hypothesis correction (Fig. 4B).

We then sought to use PACER as a tool to identify whether clonal expansion represented the underlying mechanism for any other loci previously identified associated with mCA prevalence. We interrogated loci identified in our previously reported GWAS of expanded mCA clone size, where expanded clones were defined as clonal fraction > 10% of blood. In addition to SNPs overlapping *TCL1A*, we identified

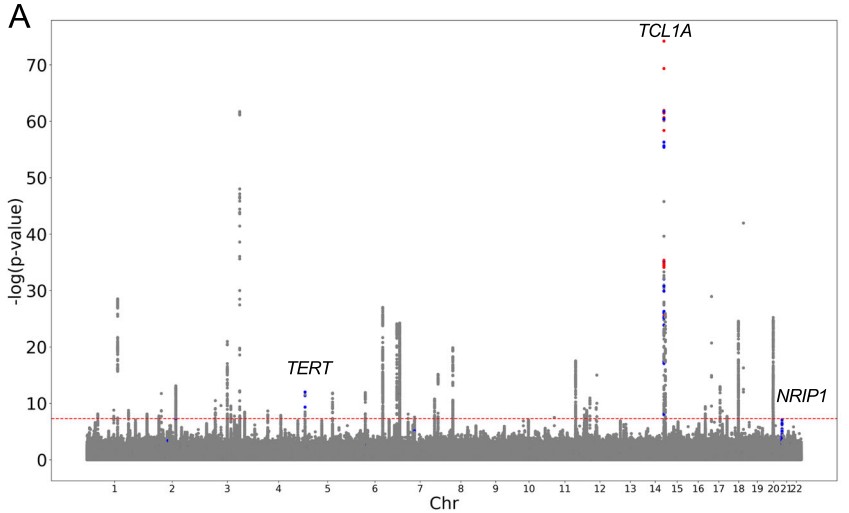

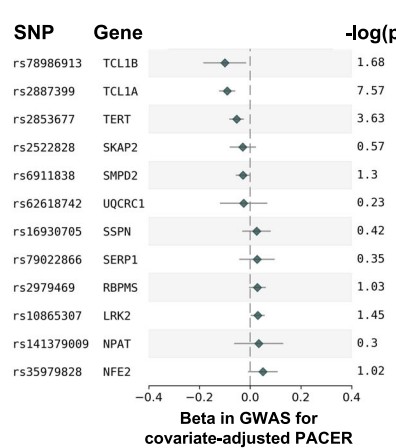

**Fig. 5 | Replication of genome-wide association study of PACER score with prior expanded clone genome-wide association study. A** Manhattan plot from the GWAS in Zekavat et al. for expanded clonal size (defined as clonal fraction > 10%) among N = 444,199 individuals in the UK Biobank, of which N = 66,011 carried an mCA and N = 12,398 individuals carried an expanded clone. The GWAS in Zekavat et al was performed with a Wald logistic regression model with covariate-adjustment for age, age[2], sex, ever smoking, principal components 1–10, and genotyping array. Red points represent SNPs in the expanded clonal size GWAS with a p-value < 10⁻⁶ in the GWAS for passenger-approximated clonal expansion rate, and blue points represent SNPs in the expanded clonal size GWAS with a p-value < 10⁻³

and > 10⁻⁶ in the GWAS for passenger-approximated clonal expansion rate. These included SNPs in the genes *TCL1A* (chr14), *TERT* and *NRIP1* (chr21). **B** Forest plot of the most significant SNPs in the GWAS for expanded clonal size demonstrating their coefficients and -log(p-value) in the GWAS of passenger-approximated clonal expansion rate. The effect estimates were derived from a linear mixed model with kinship adjustment regressing inverse normally transformed total passenger mutations, with age, sex, clonal fraction, study, and the first 10 ancestral principal components included in the model as covariates. Genes annotated based on OpenTargets variant annotations. SNPs with beta < 0.025 not shown. Data are presented as mean values ±1.96*SE.

that SNPs overlapping *NRIP1* and *TERT* had a p-value less than $1 \times 10^{-3}$ in our mCA PACER analysis (Fig. 5A). Among the reported hits in the clonal expansion GWAS, both rs2887399 in *TCL1A* and rs2853677 in *TERT* were negatively associated with total passenger mutation count in the PACER score GWAS (Fig. 5B).

To identify rare germline variants associated with PACER score among individuals with mCAs, we performed combined burden and variance-component tests implemented in REGENIE. We restricted our analysis to rare variants in coding regions of autosomal chromosomes (see Methods). We tested for associations with 17,668 genes, using five annotation masks, four of which predicted loss of function. No genes were significantly associated with mCAs (Supplementary Fig. 4 and Supplementary Table 1).

## Discussion

CH, often caused by SNVs or mCAs, is a common age-related condition that increases risk for hematologic malignancy, cardiovascular disease, liver disease, and all-cause mortality[3,10,11,22–24]. The factors underlying clonal expansion in CH caused by mCAs have not been studied and are poorly understood. Here, we estimated the rate of clonal expansion in individuals with gain, loss, and CN-LOH mCAs in TOPMed. The PACER method has been previously validated in SNVs that cause clonal hematopoiesis of indeterminate potential (CHIP)[20]. Here, we extend the approach to a distinct form of somatic mosaicism highlighting the generalizability of the approach and permitting several observations.

The most observed mCAs involved chromosomes 1, 9, and 11—specifically CN-LOH in these chromosomes. The likely explanation for the increased prevalence of these mCAs is that these chromosomes contain the genes *MPL*, *JAK2*, and *ATM* respectively and somatic mutations in these genes make these mCAs more detectable due to their increased proliferative ability. Across mCA types in TOPMed, CN-LOH mutations were the most common, likely due to the increased ability to detect CN-LOH events due to their larger size and their effect on both haplotypes.

First, not only did the mCA clonal expansion rate vary significantly across chromosomes but also among individuals with the same mCA chromosome and type, highlighting that the mCA mutation was an incomplete determinant of clonal fitness. When we aggregate our PACER scores by mCA location and type to estimate mCA fitness, the fitness estimated from passenger mutations correlated with an orthogonal approach by Watson and Blundell in the UK Biobank. Watson and Blundell use an evolutionary model of HSC dynamics to estimate the fitness of mCAs based on the clonal fraction distribution for individuals with the mCA[18]. However, our method uniquely estimates the expansion rate for a clone within an individual, enabling single-variant analysis to find germline risk factors of clonal expansion and associations of fitness with clinical phenotypes, such as peripheral blood counts. The correlation observed between passenger-estimated fitness and CF-derived fitness suggests that PACER can provide per-individual fitness estimates comparable to fitness estimated with population-derived methods.

Second, the fitness of mCAs estimated with PACER was lower than that of somatic SNV mutations in leukemia driver genes that cause CHIP. While CHIP mutations affect a single gene, mCAs typically span large genetic regions affecting the gene dose of dozens or even hundreds of genes. It is likely that these gene sets confer a mixture of both selective advantages balanced by deleterious consequences on the clonal outgrowth leading to overall decreased fitness compared to single gene mutations.

Third, we identified specific germline genetic determinants that contribute to an individual's mCA expansion rate. We performed the first ever GWAS to find germline associations with mCA clone expansion rate. The GWAS, which identified *TCL1A* locus at genome-wide significance, suggests aberrant activation of TCL1A may also promote clonal expansion for mCAs as it does for CHIP. TCL1A is known to be part of the PI3K-Akt-mTOR signaling pathway. Previous work has proposed that the acquisition of a driver mutation can increase the accessibility of the pro-proliferative *TCL1A* gene, which promotes clonal expansion[25]. This observation is convergent with prior

observations that somatic rearrangements in *TCL1A* are implicated in lymphoid malignancies (specifically T-prolymphocytic leukemia)[20,26,27]. The results of this study provide further support for the role of TCL1A in clonal expansion.

We also leveraged our mCA expansion GWAS to perform sensitivity analyses of prior GWASes of expanded clone size. We identify NRIP1 as modulating mCA prevalence by affecting clonal expansion rate. NRIP1 is a regulator of oncogenic signaling pathways in chronic lymphocytic leukemia and a therapeutic target to sensitize acute myeloid leukemia to all-trans retinoic acid[28,29]. Alternative SNPs in the *NRIP1* gene have been previously associated with increased WBC and monocyte count[30]. TERT is a ribonucleoprotein polymerase that maintains telomere ends by addition of the TTAGGG repeat; its dysregulation in somatic cells is associated with oncogenesis. Rare variant association tests did not identify additional modulators of mCA clone expansion rate. Thus, our genetic analyses suggest clonal expansion as a putative mechanism for selected loci associated with expanded mCA clone size.

Fourth, we find that mCA clonal expansion rate has phenotypic consequences. Previous work has identified an association between mCAs, white blood cell counts and infection rate, indicating that clonal expansion may lead to decreased ability to fight infection[17]. We observe that faster mCA clonal expansion rate—higher PACER score—was associated with increased measured erythrocyte counts among individuals with mCAs or somatic single nucleotide variants affecting *JAK2*. As the presence of an elevated erythrocyte count in combination with mutations in *JAK2* suggest a diagnosis of polycythemia vera, our observation suggests that mCA expansion rate in clones with lymphoid driver mutations may have utility in prognosticating risk of progression to hematologic malignancy[31].

While our study provides novel insights into the fitness and germline modulators of mCAs, it has several limitations. First, TOPMed has a limited sample size of individuals with mCAs (6381 individuals), so most mCAs were not present in more than 25 individuals and thereby were underpowered. Second, we were only able to study individuals with a single mCA due to limitations of PACER. Nonetheless, only approximately 9% of individuals with mCAs in TOPMed had multiple, so PACER is still applicable to most individuals with mCAs. Third, our results suggest that for some outlier mCAs, passenger mutations may be overestimated on the chromosome with the mCA due to structural rearrangements; however, this limitation does not seem to meaningfully affect our results. Fourth, we do not currently have serial samples of measured mCA clonal fraction, so we are currently unable to validate our estimates of expansion rate. However, PACER estimates of clonal expansion have been successfully validated in individuals with SNV driver mutations[20].

In summary, leveraging the per-individual mCA clonal expansion rate estimates from PACER, we compared aggregate fitness of different mCA types and locations, identified germline determinants of mCA expansion rate and phenotypic consequences. These findings highlight potential treatment targets for mCA expansion rate and provide an approach to identify individuals at the highest risk of mCA-driven disease progression.

## Methods

### Study samples
For this study, we leveraged the NHLBI Trans-Omics for Precision Medicine (TOPMed) dataset, which has whole-genome sequencing (WGS) on 127,946 samples from 51 studies with informed consent. The characteristics of this sample have been previously described[20]. The study design was approved by the Vanderbilt Institutional Review Board (IRB#210270).

### Identification of mosaic chromosomal alterations with MoChA
Using WGS data from 67,390 individuals in TOPMed, we identified 7693 individuals with mosaic chromosomal alterations using MoChA

version 1.11[21]. MoChA relies on haplotype-phasing to detect mCAs. Haplotype phasing was performed with Eagle 2.4 in NHLBI's TOPMed Informatics Research Center (IRC)[32]. Using these phased genotypes, MoChA evaluates coverage and B allele frequency (BAF) at heterozygous loci to detect mCAs. Heterozygous markers from Taliun et al were used[33]. The MoChA tool was executed with the additional parameter '−LRR-weight 0.0−bdev-LRR-BAF 6.0', which deactivated the LRR + BAF model. MoChA is a method that identifies mCAs to find mCA-induced deviations in allelic balance at heterozygous sites[5,13]. An mCA was defined as a gain, loss, or copy-neutral loss of heterozygosity in a specific chromosome and p or q arm. Code is available at https://github.com/freeseek/mocha.

We excluded 160 samples with phased BAF auto-correlation >0.05, indicative of contamination or other potential sources of poor DNA quality, and 67 samples with phenotype-genotype sex discordance. We removed likely germline copy number polymorphisms (lod_baf_phase <20 for autosomal variants and lod_baf_phase <5 for sex chromosome variants), constitutional or inborn duplications (mCAs 2–10 Mb with relative coverage >2.25, and mCAs 50–250 Mb with relative coverage >2.5) and deletions (filtering out mCAs with relative coverage <0.5). We defined a threshold of minimum mCA size at 2 Mb and excluded mCAs with size of <2 Mb. We defined mosaic loss of the X chromosome (X-) as a loss of a segment of chromosome X > 100 Mb in size and with a relative coverage <2.5. Of those individuals, 6930 people had a single mCA and 763 with multiple mCAs.

### Whole genome processing, variant calling, and exclusion of individuals with CHIP and multiple mCAs
We were able to call somatic singletons by identifying somatic SNVs that appeared in individuals with mCAs[20]. Variants with a depth below 25 or above 100 were excluded, along with variants with a variant allele frequency exceeding 35% to exclude germline mutations. Individuals with mutations in genes associated with clonal hematopoiesis of indeterminate potential (e.g., *DNMT3A, ASXL1, TET2, JAK2*) were excluded from analyses of passenger-approximated clonal expansion rate, as were individuals with multiple mCAs. The calls of somatic singletons for clonal hematopoiesis were made from the publicly available data from Bick et al, 2020[3]. Individuals with *JAK2* V617F CHIP mutations were also determined with this dataset. The total number of passenger mutations – C > T or T > C base pair substitutions – was calculated for each patient with a single mCA.

### PACER
The PACER method[20], leverages whole genome sequencing data to estimate CH clonal expansion rate from a single blood draw. Since HSCs acquire neutral passenger mutations, defined as age-associated clock-like C > T and T > C base-pair substitutions[34], at a fairly consistent rate across individuals[35–37], these mutations can be used as a proxy for the passage of time to approximate when a CH driver mutation was acquired. As the driver mutation clone expands, the clonal fraction of both driver and passenger mutations increases. Since the detection limit of WGS at 38x coverage is ~8–10% clonal fraction, this means that passenger mutations that occurred before the driver mutation (ancestral passengers) are more likely to be detectable than those that occurred after the driver mutation (sub-clonal passengers) because these passengers are private to subsequent divisions. For two individuals of the same age and with clones of equivalent size, the expectation is that the clone with more passengers is more fit, as it must have expanded to the same size in less time.

### Covariate adjustment and normalization of total passenger mutation counts
After the number of total passenger mutations was calculated, we fit a negative binomial regression model of age, sex, and clonal fraction to predict total passenger mutations using scikit-learn in Python. We then

performed a Yeo-Johnson inverse-normal transformation on the residuals using the SciPy package in Python 2.7.17. We then used these covariate-adjusted residuals to calculate PACER score. We also calculated a PACER score following the same process for total passenger mutations excluding the chromosome of the mCA as described above. We performed a linear regression to compare covariate-adjusted PACER from passenger mutations including and excluding the chromosome with the mCA.

### Aggregation of per-individual PACER scores to calculate PACER-derived mCA fitness

To derive the PACER-estimated mCA fitness for a given mCA chromosome and type, we computed the median of PACER scores for all individuals with a given mCA type and chromosome. We then calculated the fold change for each estimate of mCA fitness relative to loss of the X chromosome (loss of > 100 Mb segment of chromosome X) by taking the ratio of the mCA fitness and the fitness of loss of the X chromosome. Multiple linear regression was performed between clonal-fraction-derived fitness from Watson and Blundell, 2023 and PACER-estimated mCA fitness, with median age of individuals with the mCA chromosome and type as a covariate[18].

### Association between clonal expansion rate and blood counts

Peripheral counts for leukocytes, lymphocytes, neutrophils, basophils, eosinophils, monocytes, platelets, and erythrocytes were obtained from the TOPMed dataset for each individual with an mCA, along with the age of the person at the time of blood draw. The blood draw for sequencing and blood counts was within 2 years for all patients, and for 67% it was the same blood draw. Myeloid cell counts were determined by summing counts of neutrophils, basophils, eosinophils, and monocytes. We employed previously defined curated sets of mCAs known to be associated with lineage-specific hematologic malignancies[14,15]. Lymphoid mCAs included gain of chromosome 12, loss of the q arms of chromosomes 10 and 13, and CN-LOH of the q arms of chromosomes 8, 9 and 13. Myeloid mCAs included loss of the q arms of chromosomes 20 and 5, gain of chromosome 8, and CN-LOH of the q arms of chromosomes 9, 14, and 22. mCAs associated with polycythemia vera were defined as CN-LOH or loss of the p arm of chromosome 9. We used a two-tailed t-test to assess for differences in lymphocyte counts between lymphoid mCAs and all other mCAs and myeloid cell counts between myeloid mCAs and all other mCAs. We then used ordinary least squares regression to perform a multiple regression of age at time of blood draw, sex, clonal fraction, and PACER to predict lymphocyte count among individuals with lymphoid mCAs, myeloid cell counts for individuals with myeloid mCAs, and erythrocyte counts for individuals with mCAs associated with polycythemia vera.

### Single variant association

Single variant association for each variant with minor allele frequency greater than 1% in individuals with a single mCA was performed with SAIGE[38,39]. Analysis was performed using the TOPMed Encore analysis server (https://encore.sph.umich.edu). Covariates in the model were age at blood draw, sex, clonal fraction, TOPMed study, and the first ten genetic ancestry principal components. We applied an inverse normal transformation to the passenger counts. We declared variants from this analysis as significant if their p-value was less than $5 \times 10^{-8}$.

### Linkage disequilibrium and conditional analysis for rs1122138

To determine whether in *TCL1A* rs1122138 is a distinct signal from rs2887399, a previously reported SNP associated with clonal expansion of SNV CH[20], we used the LDpair tool on LDLink (https://ldlink.nci.nih.gov). The $R^2$ was used to assess for linkage disequilibrium. Then, we used PLINK to perform a conditional analysis to assess whether the association signal at rs1122138 remains significant after adjusting for the effect of rs2887399[40].

### Rare single variant association

For the rare variant analysis, the omnibus test SKATO was selected because it combines variance component tests and burden tests. This analysis was implemented using a Regenie v3.2 pipeline, using the docker image released by the software creators[41]. The covariates for steps 1 and 2 were age at blood draw, inferred sex, and the first ten principal components. Step 1 was restricted to a random selection of 500,000 extremely common variants. Step 2 variants were rare (MAF < 0.01), in coding regions with mask annotations: nonsynonymous, stop-gain, stoploss, splicing, and exonic. The Bonferroni corrected significance threshold was $0.05/102140 \approx 4.09 \times 10^{-7}$.

### Reporting summary

Further information on research design is available in the Nature Portfolio Reporting Summary linked to this article.

## Data availability

Individual whole-genome sequence data for TOPMed whole genomes, individual-level harmonized phenotypes and the CHIP variant call sets used in this analysis are available through restricted access via the dbGaP TOPMed Exchange Area available to TOPMed investigators. Data for each participating study can be accessed through dbGaP with the corresponding TOPMed accession numbers: Amish (phs000956), ARIC (phs001211), BioMe (phs001644), BAGS (phs001143), CARDIA (phs001612), CFS (phs000954), CHS (phs001368), COPDGene (phs000951), FHS (phs000974), GeneSTAR (phs001218), GENOA (phs001345), GOLDN (phs001359), HCHS/SOL (phs001395), HyperGEN (phs001293), JHS (phs000964), MESA (phs001416), VU_AF (phs001032), WGHS (phs001040) and WHI (phs001237). Controlled-access release to the general scientific community via dbGaP is ongoing. GWAS summary statistics for PACER amongst individuals with mCAs are available in the following GitHub repository here: https://github.com/bicklab/pacer-mca-fitness/blob/main/Data/gwas_results_pacerint_nochip_df.assoc.

## Code availability

The code to call mosaic chromosomal alterations with MoChA is available here: https://github.com/freeseek/mocha. Passenger count variant calling pipeline is available here: https://github.com/weinstockj/passenger_count_variant_calling. Code for the analyses of this paper can be found here: https://github.com/bicklab/pacer-mca-fitness.

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

## Acknowledgements

We thank the studies and participants who provided biological samples and data for the NHLBI TOPMed Consortium. WGS for the Trans-Omics in Precision Medicine (TOPMed) program was supported by the National Heart, Lung, and Blood Institute (NHLBI). Centralized read mapping and genotype calling, along with variant quality metrics and filtering, were provided by the TOPMed Informatics Research Center (3R01HL-117626-02S1; contract HHSN268201800002I). Phenotype harmonization, data management, sample identity quality control, and general study coordination were provided by the TOPMed Data Coordinating Center (R01HL-120393; U01HL-120393; contract HHSN268201800001I). The full study-specific acknowledgments are included in the Supplementary Materials. The views expressed in this manuscript are those of the authors and do not necessarily represent the views of the National Heart, Lung, and Blood Institute; the National Institutes of Health; or the U.S. Department of Health and Human Services. We wish to acknowledge the contributions of the consortium working on the development of the NHLBI BioData Catalyst ecosystem. This work was supported by National Institutes of Health grant R01AG083736 (A.G.B., P.L.A., P.A.S.), National Institutes of Health grant R01HL117626, National Institutes of Health contract HHSN268201800002I, National Institutes of Health grant R01HL120393, National Institutes of Health grant U01HL120393, National Institutes of Health contract HHSN268201800001I, National Institutes of Health grant DP5OD029586 (A.G.B.), Burroughs Wellcome Foundation Career Award for Medical Scientists (A.G.B. and S.J.), NHLBI BioData Catalyst Fellowship (J.S.W.), and National Institutes of Health grant T32GM007347 (Y.P.).

## Author contributions

These authors contributed equally: Yash Pershad and Taralynn Mack. Y.P., T.M, A.G.B., A.R., P.L.A., P.S., S.J. and J.S.W. conceived of the study. A.G.B., J.S.W. and S.J. conceived of PACER. J.S.W. performed somatic variant calling. Y.A.J., P.L.A., A.P.R. and P.S. contributed mosaic chromosomal alteration calls in NHLBI TOPMed. Y.P. and T.M. processed the mosaic chromosomal alteration calls and somatic variant calling and phenotypic associations. Y.P., T.M. and H.P. performed the human genetic association analyses, including the sensitivity analyses and rare variant analyses. Y.P., T.M. and A.G.B. wrote the manuscript with input from all authors. A.G.B. supervised the work. A.M. Stilp, B.D.M., J.P.L., E.B., R.J.L., N. Chami, Z.W., K.B., N.P., M.F., S. Redline, B.M.P., J.C.B., A. Shojaie, E.K.S., M.H.C., J.Y., D.D., D.L., A.D.J., R.M., M.T., D.A., K.E.N., L.M.R., A.C., M.F.D., S.S. Rich, J.I.R., X.G., N.J. Cox, D.M.R., N.F., P.D., A.P.R. and the NHLBI TOPMed Consortium contributed to sequencing and phenotyping of the included NHLBI TOPMed cohorts.

## Competing interests

A.G.B. and S.J. are cofounders, equity holders, and on the scientific advisory board of TenSixteen Bio. M.H.C. has received grant support from Bayer. B.M.P. serves on the Steering Committee of the Yale Open Data Access Project funded by Johnson & Johnson. Stanford University has filed a patent application for the use of PACER to identify therapeutic targets on which S.J., A.G.B. and J.S.W. are inventors (US patent 63/141,333). The patent has been licensed to TenSixteen Bio. L.M.R. is a consultant for the NHLBI TOPMed Administrative Coordinating Center (through Westat). S.S. Rich is a consultant to Westat for NHLBI TOPMed. In the past three years, E.K.S. received grant support from Bayer and Northpond Laboratories. All other authors declare that they have no competing interests.

## Additional information

[1]Vanderbilt Genetics Institute, Vanderbilt University, Nashville, TN, USA. [2]Internal Medicine, Biomedical Informatics, University of Kentucky, Lexington, KY, USA. [3]Biostatistics, School of Public Health, University of Washington, Seattle, WA, USA. [4]Dept of Medicine, Endocrinology, Diabetes, and Nutrition, University of Maryland, Baltimore, Baltimore, MD, USA. [5]Human Genetics Center, Department of Epidemiology, Human Genetics, and Environmental Sciences, School of Public Health, The University of Texas Health Science Center at Houston, Houston, TX, USA. [6]The Charles Bronfman Institute for Personalized Medicine, Icahn School of Medicine at Mount Sinai, New York, NY, USA. [7]Division of Biomedical Informatics & Personalized Medicine, University of Colorado Anschutz, Aurora, CO, USA. [8]Department of Laboratory Medicine and Pathology, University of Minnesota Medical School, Minneapolis, MN, USA. [9]Brown Foundation Institute of Molecular Medicine, McGovern Medical School, University of Texas Health Science Center at Houston, Houston, TX, USA. [10]Division of Sleep Medicine, Harvard Medical School, Boston, MA, USA. [11]Cardiovascular Health Research Unit, Departments of Medicine, Epidemiology, and Health Systems and Population Health, University of Washington, Seattle, WA, USA. [12]Cardiovascular Health Research Unit, Department of Medicine, University of Washington, Seattle, WA, USA. [13]Biostatistics, University of Washington, Seattle, WA, USA. [14]Channing Division of Network Medicine, Brigham and Women's Hospital, Boston, MA, USA. [15]National Heart, Lung and Blood Institute, Population Sciences Branch, Framingham, MA, USA. [16]Division of Allergy and Clinical Immunology, Department of Medicine, Johns Hopkins University School of Medicine, Baltimore, MA, USA. [17]Department of Biostatistics, Bloomberg School of Public Health, Johns Hopkins University, Baltimore, MA, USA. [18]Department of Epidemiology, University of Texas M.D. Anderson Cancer Center, Houston, TX, USA. [19]Department of Epidemiology, University of North Carolina Chapel-Hill, Chapel Hill, NC, USA. [20]Department of Genetics, University of North Carolina at Chapel Hill, Chapel Hill, NC, USA. [21]Department of Medicine, University of Mississippi Medical Center, Jackson, MS, USA. [22]Department of Pathology & Laboratory Medicine, The University of Vermont Larner College of Medicine, Colchester, VT, USA. [23]Center for Public Health Genomics, University of Virginia School of Medicine, Charlottesville, VA, USA. [24]Pediatrics, Genomic Outcomes, The Institute for Translational Genomics and Population Sciences, The Lundquist Institute for Biomedical Innovation at Harbor-UCLA Medical Center, Torrance, CA, USA. [25]Division of Genetic Medicine, Department of Medicine, Vanderbilt University and Vanderbilt University Medical Center, Nashville, TN, USA. [26]Departments of Medicine, Pharmacology, and Biomedical Informatics, Vanderbilt University Medical Center, Nashville, TN, USA. [27]Weill Cornell Medical College, New York, NY, USA. [28]Public Health Sciences Division, Fred Hutchinson Cancer Center, Seattle, WA, USA. [29]Division of Biostatistics, Insitute for Health & Equity and Cancer Center, Medical College of Wisconsin, Milwaukee, WI, USA. [30]Dept of Epidemiology, University of Texas M. D. Anderson Cancer Center, Houston, TX, USA. [31]Department of Pathology, Stanford University, Stanford, CA, USA. [32]Center for Statistical Genetics, Department of Biostatistics, University of Michigan School of Public Health, Ann Arbor, MI, USA. [33]These authors contributed equally: Yash Pershad, Taralynn Mack. ✉e-mail: alexander.bick@vumc.org

