## [Peer Review File · Nature Communications]

Determinants of mosaic chromosomal alteration fitnessREVIEWER COMMENTS

Reviewer #1 (Remarks to the Author):

The manuscript titled "Determinants of mosaic chromosomal alteration fitness" by Pershad et al. employs the Passenger-Approximated Clonal Expansion Rate (PACER) method to estimate the clonal expansion rate for 6,381 individuals in the NHLBI TOPMed cohort with mosaic chromosomal alterations (mCAs) identified in whole-genome sequence data (WGS). The study reveals that individuals with lymphoid-associated mCAs have a significantly higher white blood cell count and a faster clonal expansion rate. The study also identified several locus variants that modulate the mCA clonal expansion rate. Overall, the study is well-conducted; the methods are clearly described and sound; the paper is well-written and balanced.

I have some comments:

1. The PACER method for estimating mCA fitness is unique, but can only be applied to whole-genome sequence (WGS) data. Most mCA analyses have predominantly been conducted on SNP array data. This limitation makes the method only applicable to a small number of mCA studies in practice.

2. Unlike the conventional approaches that require serial blood samples, the PACER method estimates clonal fitness from a single blood draw. However, longitudinal samples can capture clonal expansion for individuals with any number of the mCAs. The PACER method is only applicable to subjects with one mCA. This limitation further restricts its scope of use in practice, especially in hematological samples where multiple mCAs in a single individual are common.

3. Fig3a: lymphocyte counts vs clonal expansion rate: The conclusion seems to be influenced by two outliers (with counts > 10). I suggest exploring the robustness of the conclusion by repeating the same analysis, excluding these two outliers.

4. Line 166 "We aggregated the individual data by mCA type and location and calculated the fold-change in estimated clonal expansion rate relative to a loss of chromosome X"

Please clarify the meaning of "a loss of chromosome X". Specify whether it refers to females losing one copy of chrX or to those females with mosaic loss of chromosome X (mLOX), defined as mCA > 100 MB in size and rel_cov < 2.5.

5. In the methods section for MoChA:

- (i) The "three hidden Markov models" (line 341) should be the "3-state hidden Markov model."
- (ii) MoChA employed a haplotype-based detection method. Please provide the name and version of the software used for phasing.

6. Is any threshold used for the minimal mCA size in the analysis (e.g., > 2MB)?

7. There are a few instances where "driver mCA" is used instead of mCA. Please ensure consistency or define if "driver mCA" represents a subset of mCAs.

Reviewer #2 (Remarks to the Author):

The authors selected 6,381 individuals with WGS data from the NHLBI TOPMed cohort with a detectable mosaic chromosomal alteration (mCA) inferred using the MoChA software and computed the number of passenger mutations by looking for mutations with a lower than 50/50 representation of alternate and reference alleles in the sequencing data. They then adjusted this number using age, sex, and clonal fraction to assign to each individual a PACER score and compute for each mCA class

(three classes for each autosome: CN-LOH, gain, and loss) a fitness rate. This follows their previous work in Nature (Weinstock et al. 2023) where they estimated the same values for driver genes mutation classes instead of mCA classes. They then compared the mCA class fitness rates to the fitness rates from Blundell et al. 2023, which were based solely on the distribution of clonal fractions for each mCA class and showed a significant correlation of $r^2=0.49$. They then used the PACER scores as phenotypes for a quantitative GWAS within the 6,381 individuals with mCAs and identified *TCL1A* as genome-wide significant ($p=3.1e-8$). The work is an interesting and novel analysis combining mCAs detectable from integrating deviations across consecutive heterozygous sites and somatic mutation counts across the genome ascertainable through available whole genome sequencing data

Major comments:

Although the PACER scores used in this manuscript are passenger mutation counts adjusted for clonal fraction, I find somewhat hard to believe that the authors can assay well this number given that the vast majority of mCAs are at low clonal fractions and therefore passenger mutations would be very difficult to assay. Although the PACER score is adjusted for clonal fraction, it would be nice to see a scatter plot of clonal fraction vs. passenger mutations, colored or stratified by mCA super-type (loss/gain/CN-LOH) to give an idea about what is going on. I am a little bit concerned that the PACER score, although still being an informative statistic as the authors clearly show in this paper, might be something unrelated to the mCA but rather correlated to it through some third hidden variable (such as propensity for the blood to become clonal). In this sense, the claim in the abstract that variants in *TCL1A*, *NRIP1*, and *TERT* are estimates of mCA expansion rate should not be made as it cannot be proven (it could be related to clonality in general rather than having any direct relation to mCAs). As in the abstract the authors claim that mCA (class) fitness estimates were correlated ($R^2=0.49$) with CF-fitness estimates. What is the correlation of each of these two fitness estimates with median clonal fraction?

Similarly the number of detectable passenger mutations is likely to be strongly correlated with the clonal fraction. Although PACER scores are adjusted for clonal fraction, it would be interesting to see visually how the passenger mutations counts (and the PACER scores) are related to clonal fraction

The manuscript needs quite a bit of rewriting as it is very difficult to understand what statistics refer to. I would advise to make sure that terms are always defined before being used and to use consistency. Sometimes the authors use PACER to identify a method, sometimes the authors use PACER when they really meant PACER score. Sometimes they use fitness to indicate some statistics that apply to mCA type, and sometimes as some statistics that apply to individuals carrying at least one mCA. This makes reading the paper quite confusing and caused me to often have to guess at what the authors had in mind. Make clear mCA fitness and PACER scores are properly defined and what they apply to (i.e. mCA classes or individuals) before being used. The authors also should not assume the readers are overly familiar with their original PACER paper (Weinstock et al. 2023)

I find the GWAS section of the paper a bit confusing and misleading. When comparing the loci with the GWAS of expanded mCA clone size from Zekavat et al. 2021, I am worried that you are incurring an indirect comparison of determinants for mLOY. *TCL1A* is the strongest association with mLOY and both expanded clone sizes and large PACER scores would inevitably be associated with mLOY as mLOY is very prevalent and always detected at high clonal fractions compared to other mCAs. To investigate whether this is the case, the author could at least check whether the association holds similarly in males and females and, depending on what they identify, decide whether to rewrite the section. Also the authors should only include *TCL1A* as a GWAS result in the abstract as the SNPs in *TERT* and *NRIP1* are replications that could be confounded in many ways by indirect correlations. It is not correct to state that the three loci were identified through GWAS which strictly requires $p<5e-8$

I could not find in the Methods an explanation of how the mCA (class) fitness rates were computed. I assume one has to follow the Weinstock et al. 2023 paper and read the section "Fitness estimates for driver genes" but given how relevant this step is in the manuscript, it should be explained without the

reader having to follow up on this reference. It might actually be beneficial to have a summarizing diagram, maybe as a separate figure rather than Fig. 1A which does not go into any detail, explaining how, from WGS data, age, sex, and clonal fraction one goes and calculates (i) who has mCAs; (ii) how many passenger mutations individuals have; (iii) how you compute PACER scores; (iv) how you compute mCA (class) fitness; (v) how you run the GWAS

Minor Comments:

Page 3: These driver mutations can be single nucleotide variants in cancer driver genes or larger structural rearrangements called mosaic chromosomal alterations (mCAs) <- these ones are the ones we routinely detect from DNA microarrays, WES, and WGS, but there are also translocations that we do not routinely detect so it is important to specify that these are some of the drivers rather than all of the drivers. Also notice that clonality, for what we know, could be also driven by alterations of the methylation profiles that we don't routinely assay

Page 3: We used the Passenger-Approximated Clonal Expansion Rate (PACER) method to estimate clonal expansion rate for 6,381 individuals in the NHLBI TOPMed cohort with gain, loss, and copy-neutral loss of heterozygosity mCAs. Our estimates of mCA fitness were correlated ... <- this is going to be very unclear in the abstract as you have two estimates in this paper: (i) the adjusted PACER scores for each individual with an mCA and (ii) the fitness for each mCA type and if you need to make sure readers of the abstract understand this concept

Page 3: with an alternative approach that estimated fitness of mCAs in the UK Biobank using a theoretical probability distribution <- I think it would be more appropriate to say "... in the UK Biobank based on clonal fractions distributions" as their fitness is a function of those values only

Page 4: when a clone of cells is detectable without causing blood count abnormalities or hematologic malignancy <- not sure here what you mean with abnormalities. Maybe you mean changes that lead to clinical diagnosis ... there are numerous reported associations between mCAs and blood counts (see Loh, P.-R., Genovese, G. & McCarroll, S. A. Monogenic and polygenic inheritance become instruments for clonal selection. Nature 584, 136–141 (2020) and Lin, S. et al. Mosaic chromosome Y loss is associated with alterations in blood cell counts in UK Biobank men. Scientific Reports (2020))

Page 4: rather than for a mutation in an specific individual <- rather than for a mutation in a specific individual

Page 4: CH is common and increases with age <- CH is not common in young individuals so this reads weird

Page 4: Previous studies have shown that CH significantly increases an individual's risk of all-cause mortality and many common complex diseases <- many of these studies show associations, not causations, so it is incorrect that CH increases the risk

Page 4: mCAs ... occur in about 3% of individuals older than 50 years old without cancer <- the rate of detectable mCAs is much higher after age 50, even if you only restrict to autosomal mCAs (see Terao, C. et al . Chromosomal alterations among age-related haematopoietic clones in Japan. Nature (2020))

Page 4: gain or loss of an entire chromosome <- when you restrict to the autosomes, loss of an entire chromosome in blood is almost never observed (likely it is not tolerated) so this sentence could be rewritten

Page 4: Although individuals with larger mCA clones generally have worse health outcomes <- I think you mean larger cell fraction clones, but this reads as mCAs that are larger so rewrite this. Also, I think not all the references in this sentence support this claim so make sure you reference supports worse health outcomes for larger cell fractions

Page 4: so obtaining sufficient sample sizes is challenging <- this is only challenging because we are using datasets generated for reasons other than testing for CH so this needs to be better explained

Page 4: Estimating the fitness of a given driver mutation <- even in the original PACER paper the fitness was computed for mutated driver genes rather than single mutations. Although that might be the goal, I find it confusing as the fitness is always computed about a class of mutations

Page 4: to estimate the fitness of a SNV with a single blood sample <- this is incorrect as in the original PACER paper you do not estimate the fitness of a SNV but classed of SNV (binned by driver gene)

Page 4: we apply PACER to 6,381 individuals with gain, loss, and CN-LOH mCAs in the NHLBI Trans-Omics for Precision Medicine (TOPMed) dataset <- given the right assay, anybody could be classified as having gain, loss, and CN-LOH so it is important to specify that these are events detected with a given assay (WGS)

Page 4: using our individual mCA fitness estimates <- the fitness is for mCA classes ... I think what you mean here is PACER scores

Page 5: We identified 6,930 people with one mCA and 763 with multiple mCAs <- how many had no mCAs detected?

Page 5: After excluding mCAs of sex chromosomes, we detected 3,828 autosomal mCAs in 3,068 unique individuals <- how many of these individuals had detected sex chromosome mCAs?

Page 5: the minimum total passenger mutations ... was <- the minimum total passenger mutations count ... was

Page 5: We calculated a covariate-adjusted PACER <- We calculated a covariate-adjusted PACER score

Page 5: we compared passenger mutation counts between mCA chromosomes <- I understand what you mean here but you need to define first what you mean with the term mCA chromosomes

Page 5: The most common autosomal mCAs were on chromosomes 11, 1, and 9 <- I bet these are mostly 1q CN-LOHs driven by MPL, 9p CN-LOHs driven by JAK2, and 11q CN-LOHs driven by ATM. However, as the data is reported, the less experienced reader will have no way to make this determination. It would be helpful here to be a bit more specific

Page 5: with CN-LOH being the majority (52%). Clonal fraction showed significant variation across autosomal mCA types and chromosomes (Fig 1c) <- A lot of reader will not understand that this is at least partially explained by CN-LOHs being much easier to detect both because of their larger size and because they affect two haplotypes at once. Similarly gains are easier to detect than losses as they tend to be larger. It must be made clear that cell fraction differences are mostly driven by limits of detection rather than by differences in biological processes. It's also not clear how by looking at Fig 1c you can assess that there is significant variation in clonal fractions

Page 5: we quantified mCA fitness in individuals with a single mCA and no CH-associated SNVs <- my understanding is that fitness is a statistic assigned to a class of mCAs, not to an individual ... there is quite some confusion in the writing

Page 5: the minimum total passenger mutations for a patient with an mCA was <- at this point passenger mutations are not defined. Although they are then later defined in the methods, there is no "see Methods" notice anywhere before explaining that there is a definition later on. As I was reading the paper I had no idea what a passenger mutation was, especially given the very specific definition

Page 5: We calculated a covariate-adjusted PACER <- before this instance PACER is described as a method while here it seems to be a statistic or a score ... this is confusing. Again, make sure everything is properly defined before being used. Do you mean here a statistic being equal to the PACER fitness of the mCA carried by the individual after adjusting for covariates?

Page 5: As a sensitivity analysis <- Maybe you meant something like "to check for potential confounders"

Page 5: Furthermore, we found that PACER scores <- notice here you use PACER score but this is never defined

Page 5: than driver SNVs in non-R882 DNMT3A <- why choose this one from the PACER paper? Maybe explain the rationale behind this

Page 5: A generalized linear model of median PACER <- is this the median of the PACER scores across individuals carrying an mCA of the same mCA class?

Page 6: , individuals with lymphoid mCAs within TOPMed had a significantly ($p = 0.0004$) higher lymphocyte count ... than those with non-lymphoid mCAs <- report here the number of individuals with lymphoid mCAs and those with non-lymphoid mCAs. Also, how did you count the individuals that had both?

Page 6: Among individuals with lymphoid mCAs, a multiple regression of age at time of blood draw, sex, clonal fraction, and PACER predicting lymphocyte count explained 13.3% of the variance in lymphocyte counts, compared to only 2.5% for a model without PACER (Fig 3a) <- by the figure it seems as if this might be driven by a few outliers ... what is the p-value associated with this increase? (I see this is reported later as $p=0.019$... maybe just incorporate the two sentences so that it is clear

that the statement is only nominally significant)

Page 10: MoChA is a method that identifies mCAs using three hidden Markov models to find mCA-induced deviations in allelic balance at heterozygous sites <- this is incorrect as stated at <https://github.com/freeseek/mocha#hmm-parameters> where it says "There are a few differences between MoChA and the HMM model used in Loh et al. 2018, Thompson et al. 2019, Loh et al. 2020, and Terao et al. 2020. The most important is that, the latter model used a likelihood ratio test statistic with likelihoods deriving from a 3-state forward-backward HMM model. MoChA instead uses a Viterbi HMM model with multiple alternate states and it increases the number of alternate states dynamically when trying to assess multiple calls". MoChA tests a large array of different cell fractions as multiple hidden states all at once. Although essentially similar to what done in the original studies, it does not run a three states HMM

Page 10: As was done in Weinstock et al, passenger mutations were defined as C>T and T>C base-pair substitutions that do not occur in a CH-associated gene. These mutations are "clock-like" and are associated with aging, whereas other mutations may be affected by disease or environmental factors <- is this because most C>T and T>C base-pair substitutions are CpG mutations? If so, was there a reason why you didn't simply look at CpG mutations (i.e. transitions at CpG sites)?

Page 11: Linkage disequilibrium and conditional analysis for rs1122138 <- explain in this section that these are two variants in the TCL1A gene

Page 13: GWAS summary statistics for PACER amongst individuals with mCAs are available in the following GitHub repository here: https://github.com/bicklab/pacer-mca-fitness/blob/main/Data/gwas_results_pacerint_nochip_df.assoc <- I assume this will become available once the paper is published as at the moment there is no pacer-mca-fitness repository

Page 15: Scatter plot of median total passenger mutations and fitness derived from clonal fraction by mCA <- this is a bit confusing as the x-axis is CF-derived fitness, not median total passenger mutations ... I am confused about what is actually being plotted here

Page 15: only includes mCAs with counts > 25 <- what is counts? I assume this is the number of individuals with a given mCA type, so say so and don't interchangeably use mCA and mCA type as it is hard to understand what you are plotting here ... mCA is one observed event, not a class of events

Page 23: Zekavat, S. M. et al. Hematopoietic mosaic chromosomal alterations and risk for infection among 767,891 individuals without blood cancer. medRxiv 2020.11.12.20230821 (2020)

doi:10.1101/2020.11.12.20230821.599 <- this preprint has now been published in Nature M

Page 23: Watson, C. J. & Blundell, J. R. Mutation rates and fitness consequences of mosaic chromosomal alterations in blood. 2022.05.07.491016 Preprint at

<https://doi.org/10.1101/2022.05.07.491016> (2022) <- this preprint has now been published in Nature Genetics

Figure 1b: please consider a pileup plot or a circos plot or anything that can readily distinguish mCAs by size and by chromosome arm as a histogram leaves a lot of information out (Extended Data Fig. 1a,b of Zekavat et al. 2021 is an example)

Figure 2b: the Y-axis is labeled as "fold change" but should it really be log fold-change? And what log? Natural, 2, or 10?

Extended Data Fig 2: Covariate-adjusted PACER score <- notice here you use PACER score but this is never defined

Reviewer #1 (Remarks to the Author):

The manuscript titled “Determinants of mosaic chromosomal alteration fitness” by Pershad et al. employs the Passenger-Approximated Clonal Expansion Rate (PACER) method to estimate the clonal expansion rate for 6,381 individuals in the NHLBI TOPMed cohort with mosaic chromosomal alterations (mCAs) identified in whole-genome sequence data (WGS). The study reveals that individuals with lymphoid-associated mCAs have a significantly higher white blood cell count and a faster clonal expansion rate. The study also identified several locus variants that modulate the mCA clonal expansion rate. Overall, the study is well-conducted; the methods are clearly described and sound; the paper is well-written and balanced.

We are appreciative of Reviewer 1 for their appraisal of our work and their helpful comments, which have benefited our manuscript. We particularly thank the reviewer for improving the precision of our wording in the methods regarding mCA detection and the data we present in Figure 3 correlating PACER score with peripheral blood counts.

We detail our response to each of the comments below. Reviewer comments are in black and the authors’ responses are in blue. Updates to the text are in blue underlined and in italics.

I have some comments:

1. The PACER method for estimating mCA fitness is unique, but can only be applied to whole-genome sequence (WGS) data. Most mCA analyses have predominantly been conducted on SNP array data. This limitation makes the method only applicable to a small number of mCA studies in practice.

Thank you for this excellent point. We agree that, to date, mCAs have been primarily called with genotyping arrays, and the reviewer is correct that the PACER method requires WGS data. In this study, our mCA calls in TOPMed were derived from the “Freeze 8” WGS data, as described in our methods and Jakubek et al, 2023 (reference below). As more individuals undergo WGS, the broader research community will be able to use these methods to call mCAs and estimate per-individual clonal expansion rate from a single WGS sample with the PACER method. We feel this makes the PACER method unique and important to understand mCA clonal expansion as WGS becomes more ubiquitous.

Reference: Jakubek, Y. A. et al. Mosaic chromosomal alterations in blood across ancestries using whole-genome sequencing. *Nat. Genet.* 55, 1912–1919 (2023).

2. Unlike the conventional approaches that require serial blood samples, the PACER method estimates clonal fitness from a single blood draw. However, longitudinal samples can capture clonal expansion for individuals with any number of the mCAs. The PACER method is only applicable to subjects with one mCA. This limitation further restricts its scope of use in practice, especially in hematological samples where multiple mCAs in a single individual are common.

The reviewer is 100% correct here. We acknowledge the limitations of the PACER method for only estimating the clonal fitness for individuals with only a single mCA; this limitation was initially described

in depth in Weinstock et al 2023 (reference below). We mention this limitation in our Discussion section and our Methods sections. Nonetheless, even among individuals with a single mCA and no mutations in known driver genes for clonal hematopoiesis of indeterminate potential (CHIP), we still have power to make biologically important discoveries as we demonstrate in our associations of PACER score with white blood cell counts among individuals with lymphoid-associated mCAs and the significant loci in our genome-wide association study of PACER score.

Out of 67,390 people with WGS data assessed for mCAs in TOPMed, 6,930 people had one mCA and 763 had multiple mCAs. Therefore, ~9% of people with any mCAs had multiple, suggesting that PACER is still applicable for most individuals with mCAs.

The discussion section now includes this sentence: “Nonetheless, only approximately 9% of individuals with mCAs in TOPMed had multiple, so PACER is still applicable to most individuals with mCAs.”

Reference: Weinstock, J. S. et al. Aberrant activation of *TCL1A* promotes stem cell expansion in clonal haematopoiesis. *Nature* 616, 755–763 (2023).

3. Fig3a: lymphocyte counts vs clonal expansion rate: The conclusion seems to be influenced by two outliers (with counts > 10). I suggest exploring the robustness of the conclusion by repeating the same analysis, excluding these two outliers.

We thank the reviewer for bringing up this excellent point. As you point out, when we exclude individuals with lymphocyte counts greater than 10, the PACER score is no longer significantly associated with lymphocyte count among individuals with lymphoid mCAs. Given the limited number of individuals with complete blood counts available in TOPMed, it is likely we are underpowered to detect a correlation between PACER score and blood counts. However, given the strong correlation between PACER scores and erythrocyte counts among individuals with copy-neutral loss of heterozygosity or loss of the p arm of chromosome 9 (9p= and 9p- respectively), we further investigated associations between PACER score and erythrocytes to validate and understand the clinical significance of the approximated expansion rate from passenger mutation count.

To do this, we compared the association between PACER score and erythrocyte counts among individuals with 9p= and 9p- mCAs to the same association among individuals with JAK2 V617F clonal hematopoiesis of indeterminate potential (CHIP). To do this, we performed multiple linear regression of age at time of blood draw, sex, clonal fraction, and PACER to predict erythrocyte count among individuals with JAK2 V617F CHIP in the same manner done for mCAs associated with polycythemia vera (9p= and 9p-). We observed a significant association ($\beta = 0.341$, 95% CI [0.133, 0.537], $p = 0.001$), and the model with PACER score included had an R^2 of 0.90 while a model with only age and sex had an R^2 of 0.32. We now have included this plot in Figure 3. These associations include individuals with either a detected mCA or detected JAK2 V617F CHIP, as we only apply PACER scores to individuals with a single clonal driver.

We believe this plot demonstrates how PACER scores reflect the phenotypic consequences of mCA and CHIP clonal expansion rate for erythrocyte counts. We therefore have updated Figure 3 to show a scatterplot of erythrocyte counts (10¹² cells/L) versus covariate-adjusted passenger associated clonal expansion rate (PACER) score for (a) patients with copy-neutral loss of heterozygosity or loss of the p arm of chromosome 9 and (b) patients with *JAK2* V617F clonal hematopoiesis of indeterminate potential.

Accordingly, we added the following to our methods to describe where we derived these *JAK2* V617F calls: “The calls of somatic singletons for clonal hematopoiesis were made from the publicly available data from Bick et al, 2020. Individuals with *JAK2* V617F CHIP mutations were also determined with this dataset.”

Therefore, we have deemphasized the finding of the correlation of PACER score and lymphocyte count among individuals with lymphoid-cancer-associated mCAs and instead replaced it with the correlation of PACER score and erythrocyte count among individuals with mCAs affecting chromosome 9 and with *JAK2* V617F CHIP.

The new relevant results section now reads:

“For the 23 individuals with lymphoid mCAs who had lymphocyte counts, a multiple regression of age at time of blood draw, sex, clonal fraction, and PACER score predicting lymphocyte count, demonstrated a significant association between PACER score and lymphocyte counts (\$\beta = 0.0175\$, 95% CI [0.003, 0.032], \$p = 0.019\$ ). However, after excluding outliers with lymphocyte count greater than \$10 \times 10^9\$ cells/L, there was no significant association. These outliers may represent elevated lymphocyte counts in three individuals with a high PACER score due to rapidly expanding mCA clones...

Since mutations in *JAK2*, such as *JAK2* V617F, on chromosome 9p are known to cause polycythemia vera, we performed a multiple regression of age at time of blood draw, sex, and PACER score to predict erythrocyte counts for 11 individuals with CN-LOH or loss of chromosome 9p and erythrocyte count. This

model explained 91.6% of the variance in erythrocyte count, and PACER score had a significant positive association with erythrocyte count ($\beta = 0.0119$, 95% CI [0.006, 0.018], $p = 0.018$) (Fig 3a).

We then examined the association between PACER score and erythrocyte counts among individuals with JAK2 V617F clonal hematopoiesis of indeterminate potential (CHIP). We performed multiple linear regression of age at time of blood draw, sex, clonal fraction, and PACER to predict erythrocyte count among individuals with JAK2 V617F CHIP and observed a significant association ($\beta = 0.341$, 95% CI [0.133, 0.537], $p = 0.001$), and the model with PACER score included had an R2 of 0.90 while a model with only age and sex had an R2 of 0.32. Therefore, for individuals with mCAs or somatic SNVs affecting JAK2, erythrocyte count is associated strongly with PACER score.”

4. Line 166 “We aggregated the individual data by mCA type and location and calculated the fold-change in estimated clonal expansion rate relative to a loss of chromosome X”

Please clarify the meaning of “a loss of chromosome X”. Specify whether it refers to females losing one copy of chrX or to those females with mosaic loss of chromosome X (mLOX), defined as mCA > 100 MB in size and rel_cov < 2.5.

We thank the reviewer for bringing up this important clarification. We used the MoChA pipeline to call mLOX; this approach of using long-range haplotype phasing can improve sensitivity in detection of large mosaic events with low cell fractions such as mLOX. Specifically, we define mLOX as a loss of a segment of the X chromosome > 100MB in size and relative coverage estimate from LRR or sequencing coverage < 2.5.

Accordingly, our methods section now reads “We defined mosaic loss of the X chromosome (X-) as a loss of a segment of chromosome X > 100 MB in size and with a relative coverage < 2.5.”

We also added more detail to our results section which now reads: “We aggregated the individual data by mCA type and location and calculated the fold-change in estimated clonal expansion rate relative to a loss of chromosome X, defined as a loss of a > 100 Mb segment of chromosome X (Fig 2b).”

5. In the methods section for MoChA:

- (i) The “three hidden Markov models” (line 341) should be the “3-state hidden Markov model.”
- (ii) MoChA employed a haplotype-based detection method. Please provide the name and version of the software used for phasing.

Thank you for these comments about MoChA - we added more details in our methods section to be more complete and precise.

(i) Based on comments from another reviewer, we have recognized an error in the methods – the version of MoChA used in this paper actually does not use a hidden markov model, so we have removed this reference. Our apologies for this error.

(ii) The phasing performed in this study was done with Eagle 2.4 (Loh et al, 2016). This phasing was performed by the TOPMed Informatics Research Center (IRC) in their “Freeze 8” release of WGS data.

Here is the new relevant section of the methods: “Using WGS data from 67,390 individuals in TOPMed, we identified 7,693 individuals with mosaic chromosomal alterations using MoChA version 1.11.21 MoChA relies on haplotype-phasing to detect mCAs. Haplotype phasing was performed with Eagle 2.4 in NHLBI’s TOPMed Informatics Research Center (IRC). Using these phased genotypes, MoChA evaluates coverage and B allele frequency (BAF) at heterozygous loci to detect mCAs. Heterozygous markers from Taliun et al were used.³³ The MoChA tool was executed with the additional parameter ‘-LRR-weight 0.0-bdev-LRR-BAF 6.0’, which deactivated the LRR + BAF model. MoChA is a method that identifies mCAs to find mCA-induced deviations in allelic balance at heterozygous sites, as described previously.^{5,13} An mCA was defined as a gain, loss, or copy-neutral loss of heterozygosity in a specific chromosome and p or q arm. Code is available at <https://github.com/freesek/mocha>.”

Reference for Eagle2: Loh PR, et al. Reference-based phasing using the Haplotype Reference Consortium panel. Nat Genet. 2016 Nov;48(11):1443-1448.

6. Is any threshold used for the minimal mCA size in the analysis (e.g., > 2MB)?

The threshold for minimal mCA size in this analysis was set at 2 megabases (Mb). Our methods now reflect this: “We defined a threshold of minimum mCA size at 2 Mb and excluded mCAs with size of < 2 Mb.”

In the literature, studies have used from 1-5 Mb, so we decided on a value within this range. The main rationale for setting a minimal mCA size is to ensure that inherited segmental duplications are not mistakenly called gain of chromosome mCAs.

Here is a histogram of mCA size for all mCAs detected on autosomal chromosomes:

We also point the reviewer to Fig. 1 of Jakubek et al, 2023 for the genomic distribution/pile-ups of autosomal mCAs in TOPMed called from WGS data. We used these autosomal calls for our analysis.

Reference: Jakubek, Y. A. et al. Mosaic chromosomal alterations in blood across ancestries using whole-genome sequencing. Nat. Genet. 55, 1912–1919 (2023).

7. There are a few instances where “driver mCA” is used instead of mCA. Please ensure consistency or define if “driver mCA” represents a subset of mCAs.

Thank you for bringing this to our attention - there is no subset of driver mCAs, and this was an error on our part. We have removed all uses of the phrase “driver mCA”.

Reviewer #2 (Remarks to the Author):

The authors selected 6,381 individuals with WGS data from the NHLBI TOPMed cohort with a detectable mosaic chromosomal alteration (mCA) inferred using the MoChA software and computed the number of passenger mutations by looking for mutations with a lower than 50/50 representation of alternate and reference alleles in the sequencing data. They then adjusted this number using age, sex, and clonal fraction to assign to each individual a PACER score and compute for each mCA class (three classes for each autosome: CN-LOH, gain, and loss) a fitness rate. This follows their previous work in Nature (Weinstock et al. 2023) where they estimated the same values for driver genes mutation classes instead of mCA classes. They then compared the mCA class fitness rates to the fitness rates from Blundell et al. 2023, which were based solely on the distribution of clonal fractions for each mCA class and showed a significant correlation of $r^2=0.49$. They then used the PACER scores as phenotypes for a quantitative GWAS within the 6,381 individuals with mCAs and identified TCL1A as genome-wide significant ($p=3.1e-8$). The work is an interesting and novel analysis combining mCAs detectable from integrating deviations across consecutive heterozygous sites and somatic mutation counts across the genome ascertainable through available whole genome sequencing data

We thank reviewer 2 for their incredibly helpful and meticulous comments. We believe these comments have greatly benefited our manuscript – particularly ensuring that our GWAS findings are robust to sex stratification, improving the precision of our wording regarding fitness/expansion rate/PACER scores, and the data we present in Figure 3 correlating PACER score with peripheral blood counts.

We detail our response to each of the comments below. Reviewer comments are in black and the authors' responses are in blue. Updates to the text are in blue underlined and in italics.

Major comments:

Although the PACER scores used in this manuscript are passenger mutation counts adjusted for clonal fraction, I find somewhat hard to believe that the authors can assay well this number given that the vast majority of mCAs are at low clonal fractions and therefore passenger mutations would be very difficult to assay. Although the PACER score is adjusted for clonal fraction, it would be nice to see a scatter plot of clonal fraction vs. passenger mutations, colored or stratified by mCA super-type (loss/gain/CN-LOH) to give an idea about what is going on. I am a little bit concerned that the PACER score, although still being an informative statistic as the authors clearly show in this paper, might be something unrelated to the mCA but rather correlated to it through some third hidden variable (such as propensity for the blood to become clonal)...As in the abstract the authors claim that mCA (class) fitness estimates were correlated ($R^2=0.49$) with CF-fitness estimates. What is the correlation of each of these two fitness estimates with median clonal fraction? Similarly the number of detectable passenger mutations is likely to be strongly correlated with the clonal fraction. Although PACER scores are adjusted for clonal fraction, it would be interesting to see visually how the passenger mutations counts (and the PACER scores) are related to clonal fraction

Thank you for these comments regarding clonal fraction. We completely agree that the mCAs we observed are at low clonal fractions.

Here, we plot PACER score per individual on a scatterplot with clonal fraction.

We see 94.5% of mCAs had a clonal fraction < 0.1 . Nonetheless, only 2% of variability in PACER score is explained by variability in clonal fraction. The correlation remained the same among autosomal mCAs alone (excluding mLOX). These data support the conclusion that passenger mutations are not simply providing information captured in clonal fraction, but instead expansion of a clone. In other words, an mCA clone with a high clonal fraction may be a result of a mutation that occurred years ago and is expanding slowly or a recently occurring clone that is rapidly expanding.

We also plotted median PACER score against median clonal fraction, aggregated across mCA chromosome and mutation type (e.g., gain of chromosome 7 = 7+). We observe no correlation between median PACER score and median clonal fraction when aggregated. These data further support that the PACER score is not simply capturing expanded clone size (i.e., high clonal fraction).

Aggregated across mCA chromosome and mutation type

Finally, we did not find a significant association between clonal fraction and Watson and Blundell's clonal-fraction-derived fitness ($R^2 = 0.01$). These data suggest that even a population-based method of deriving fitness from clonal fraction is not only capturing expanded clone size (i.e., high clonal fraction).

Aggregated across mCA chromosome and mutation type

In this sense, the claim in the abstract that variants in *TCL1A*, *NRIP1*, and *TERT* are estimates of mCA expansion rate should not be made as it cannot be proven (it could be related to clonality in general rather than having any direct relation to mCAs).

Thank you for this comment. We observe that the significant variants in our genome-wide association study are associated with higher passenger mutation count when adjusted for covariates (i.e., PACER score) among individuals with a single mCA. Based on these results, we conclude that this quantity when measured from 38x WGS in individuals with mCAs is associated with clonal expansion rate of mCAs due to enrichment for ancestral passengers in expanding clones.

We have revised the statement in our abstract to make our language more precise in the final sentence; it now reads: “In a cross-sectional analysis, genome-wide association study of estimates of mCA expansion rate identified a TCL1A locus variant associated with mCA clonal expansion rate, with suggestive variants in NR1P1 and TERT.”

The manuscript needs quite a bit of rewriting as it is very difficult to understand what statistics refer to. I would advise to make sure that terms are always defined before being used and to use consistency. Sometimes the authors use PACER to identify a method, sometimes the authors use PACER when they really meant PACER score. Sometimes they use fitness to indicate some statistics that apply to mCA type, and sometimes as some statistics that apply to individuals carrying at least one mCA. This makes reading the paper quite confusing and caused me to often have to guess at what the authors had in mind. Make clear mCA fitness and PACER scores are properly defined and what they apply to (i.e. mCA classes or individuals) before being used. The authors also should not assume the readers are overly familiar with their original PACER paper (Weinstock et al. 2023)

We apologize for the confusing terminology and have adjusted the manuscript extensively to clarify these concepts. We now refer to PACER as a method, and the per-individual estimates of clonal expansion rate derived from PACER as “PACER scores” consistently. We then now consistently use “mCA fitness” to refer to the median of the PACER scores for all individuals with an mCA involving a chromosome of a specific type (e.g., loss of chr 9). This mCA fitness value aggregates individual PACER scores to provide a fitness estimate for an mCA and enables comparisons to population-level clonal-fraction-derived estimates of mCA fitness by Watson and Blundell in their 2023 paper published in Nature Genetics. We provide some examples below where we have made this change in the results and methods sections, but have also made minor wording changes throughout the manuscript to clarify these concepts and use precise language.

In the results section:

- Passenger mutation counts definition: “We calculated passenger mutation counts, representing clock-like C>T or T>C somatic mutations, in individuals with a single mCA and no CH-associated SNVs.”
- PACER score definition: “We then calculated a covariate-adjusted PACER score to approximate clonal expansion rate for the mCA of each individual from the normalized residuals of a negative binomial regression of age, sex, and clonal fraction predicting total passenger mutation count.”
- mCA fitness definition and fold-change definition: “We computed the median of PACER scores for individuals with the same mCA type and location to derive a measure of an mCA’s fitness from PACER. With this estimation of mCA fitness derived from PACER scores, we then calculated the fold-change relative to a loss of chromosome X, defined as a loss of a > 100 Mb segment of chromosome X.”

We added a section called “Aggregation of per-individual PACER scores to calculate PACER-derived mCA fitness” to clarify these concepts.

We have extensively revised the methods section to state these definitions more clearly:

- “We then performed a Yeo-Johnson inverse-normal transformation on the residuals using the SciPy package in Python 2.7.17. We then used these covariate-adjusted residuals to calculate PACER score...”
- “To derive the PACER-estimated mCA fitness for a given mCA chromosome and type, we computed the median of PACER scores for all individuals with a given mCA type and chromosome. We then calculated the fold change for each estimate of mCA fitness relative to loss of the X chromosome (loss of > 100 Mb segment of chromosome X) by taking the ratio of the mCA fitness and the fitness of loss of the X chromosome.”

Finally, we have revamped the Methods section to further elaborate on PACER as a method to not assume familiarity with the original PACER paper by Weinstock et al. Our new section on PACER reads:

“The PACER method, (Weinstock et al., Nature, 2023) leverages whole genome sequencing data to estimate CH clonal expansion rate from a single blood draw. Since HSCs acquire neutral passenger mutations, defined as age-associated “clock-like” C>T and T>C base-pair substitutions, (Alexandrov et al., Nature Genetics, 2015) at a fairly consistent rate across individuals, (Osorio et al., Cell Reports, 2018; Mitchell et al., Nature, 2022; Williams et al., Nature, 2022) these mutations can be used as a proxy for the passage of time to approximate when a CH driver mutation was acquired. As the driver mutation clone expands, the clonal fraction of both driver and passenger mutations increase. Since the detection limit of WGS at x38 coverage is ~8-10% clonal fraction, this means that passenger mutations that occurred before the driver mutation (ancestral passengers) are more likely to be detectable than those that occurred after the driver mutation (sub-clonal passengers) because these passengers are private to subsequent divisions. For two individuals of the same age and with clones of equivalent size, the expectation is that the clone with more passengers is more fit, as it must have expanded to the same size in less time.”

I find the GWAS section of the paper a bit confusing and misleading. When comparing the loci with the GWAS of expanded mCA clone size from Zekavat et al. 2021, I am worried that you are incurring an indirect comparison of determinants for mLOY. TCL1A is the strongest association with mLOY and both expanded clone sizes and large PACER scores would inevitably be associated with mLOY as mLOY is very prevalent and always detected at high clonal fractions compared to other mCAs. To investigate whether this is the case, the author could at least check whether the association holds similarly in males and females and, depending on what they identify, decide whether to rewrite the section.

Thank you for this feedback on our GWAS results. The concern about confounding due to mLOY is an excellent one. In the submitted version of the paper, we performed a GWAS of PACER score among all individuals in TOPMed with a single mCA including mLOX (All).

To investigate your claim, we looked at overlapping mLOY calls and found that 640 out of 811 males had no mLOY clone (78.9%). Subsequently, we performed three new analyses for *TCLIA* based on your comments: 1) GWAS of PACER score in all individuals excluding males with mLOY but including males without mLOY (All - mLOY). 2) GWAS of PACER score in only females including mLOY (Females). 3) GWAS of PACER score in only males including those with mLOY, with mLOY as a covariate (Males). We plot the effect estimate with error bars representing 95% CI (1.96 * standard error) of the leading variant in *TCLIA* below.

For rs1122138, the leading variant in *TCLIA*, we observe the effect estimate remains in the same direction (approximately -0.10) for all sub-analyses, even excluding mLOY and in only Females. The large confidence interval in Males alone with mLOY clonal fraction as a covariate is likely due to reduction in sample size.

Based on these results, we conclude that the lead variant in *TCLIA* is a particularly robust result and not a result of confounding by mLOY or mLOY.

We have updated the text to state this: *“For the leading variant in *TCLIA*, we observe a statistically significant protective effect, with a negative effect estimate against PACER score, for sub-analyses excluding individuals with mosaic loss of chromosome Y and in only females.”*

Also the authors should only include *TCLIA* as a GWAS result in the abstract as the SNPs in *TERT* and *NRIP1* are replications that could be confounded in many ways by indirect correlations. It is not correct to state that the three loci were identified through GWAS which strictly requires $p < 5e-8$

Thank you for clarifying these results - our abstract now reads *“In a cross-sectional analysis, genome-wide association study of estimates of mCA expansion rate identified a *TCLIA* locus variant associated with mCA clonal expansion rate, with suggestive variants in *NRIP1* and *TERT*.”*

We have also adjusted a sentence in the discussion to read: *“We performed the first ever GWAS to find germline associations with mCA clone expansion rate.”* rather than “modulators” as it previously said.

I could not find in the Methods an explanation of how the mCA (class) fitness rates were computed. I assume one has to follow the Weinstock et al. 2023 paper and read the section "Fitness estimates for driver genes" but given how relevant this step is in the manuscript, it should be explained without the reader having to follow up on this reference. It might actually be beneficial to have a summarizing diagram, maybe as a separate figure rather than Fig. 1A which does not go into any detail, explaining how, from WGS data, age, sex, and clonal fraction one goes and calculates (i) who has mCAs; (ii) how many passenger mutations individuals have; (iii) how you compute PACER scores; (iv) how you compute mCA (class) fitness; (v) how you run the GWAS

Thank you for pointing this out. We now have a sentence in the Methods that states: *"To derive the PACER-estimated mCA fitness for a given mCA chromosome and type, we computed the median of PACER scores for all individuals with a given mCA type and chromosome."*

We thought that the suggestion to add another figure was a great way to improve clarity, so we have added an additional supplementary figure (Extended Data Fig 1) as a summarizing diagram to clarify the questions of the reviewer. The figure and its caption is below.

Extended Data Fig 1: Extended schematic of the study describing the detection of mCAs with the MoChA software and the PACER method for estimating clonal expansion rate. PACER score represents the inverse normalized residuals of a regression of total passenger mutations (C>T and T>C clock-like mutations) against covariates. To derive the PACER-estimated mCA fitness for a given mCA chromosome and type, the median of PACER scores for all individuals with a given mCA type and chromosome is computed and compared to clonal-fraction-derived fitness of mCAs by chromosome and type from Watson and Blundell 2023. VAF = variant allele fraction which is equivalent to clonal fraction.

Minor Comments:

Page 3: These driver mutations can be single nucleotide variants in cancer driver genes or larger structural rearrangements called mosaic chromosomal alterations (mCAs) <- these ones are the ones we routinely detect from DNA microarrays, WES, and WGS, but there are also translocations that we do not routinely detect so it is important to specify that these are some of the drivers rather than all of the drivers. Also notice that clonality, for what we know, could be also driven by alterations of the methylation profiles that we don't routinely assay

The introduction text now reads “Mutations in HSCs causing CH include single nucleotide variants (SNVs) in genes associated with hematological malignancies (e.g., DNMT3A, TET2, and JAK2) referred to as Clonal Hematopoiesis of Indeterminate Potential (CHIP) and larger chromosomal rearrangements called mosaic chromosomal alterations (mCAs) (Machiela et al, Am. J. Hum. Genet. 2015). This does not include non-detectable translocations or alterations to methylation profiles that can also result in clonality.”

We have also revised the Discussion section's first sentence to now say: “CH, often caused by SNVs or mCAs...” as clonality may be caused by non-detectable translocations or methylation profile alterations.

Page 3: We used the Passenger-Approximated Clonal Expansion Rate (PACER) method to estimate clonal expansion rate for 6,381 individuals in the NHLBI TOPMed cohort with gain, loss, and copy-neutral loss of heterozygosity mCAs. Our estimates of mCA fitness were correlated ... <- this is going to be very unclear in the abstract as you have two estimates in this paper: (i) the adjusted PACER scores for each individual with an mCA and (ii) the fitness for each mCA type and if you need to make sure readers of the abstract understand this concept

This point is well-taken in the context of the other confusing language in the paper, that we have edited the manuscript to resolve. Our abstract now reads: “We used the Passenger-Approximated Clonal Expansion Rate (PACER) method to estimate clonal expansion rate for 6,381 individuals in the NHLBI TOPMed cohort with gain, loss, and copy-neutral loss of heterozygosity mCAs. Our mCA fitness estimates, derived by aggregating per-individual PACER scores, were correlated...”

Page 3: with an alternative approach that estimated fitness of mCAs in the UK Biobank using a theoretical probability distribution <- I think it would be more appropriate to say "... in the UK Biobank based on clonal fractions distributions" as their fitness is a function of those values only

Thanks for this comment - the sentence in the abstract now reads “Our mCA fitness estimates, derived by aggregating per-individual PACER scores, were correlated ($R^2 = 0.49$) with an alternative approach that estimated fitness of mCAs in the UK Biobank using population-level distributions of clonal fraction.”

Page 4: when a clone of cells is detectable without causing blood count abnormalities or hematologic malignancy <- not sure here what you mean with abnormalities. Maybe you mean changes that lead to clinical diagnosis ... there are numerous reported associations between mCAs and blood counts (see Loh, P.-R., Genovese, G. & McCarroll, S. A. Monogenic and polygenic inheritance become instruments for clonal selection. Nature 584, 136–141 (2020) and Lin, S. et al. Mosaic chromosome Y loss is associated with alterations in blood cell counts in UK Biobank men. Scientific Reports (2020))

Thank you for this clarification - the sentence now reads: “The phenomenon of clonal hematopoiesis (CH) occurs when a clone of cells is detectable without causing cytopenias, dysplastic hematopoiesis, or hematologic malignancy.”

Page 4: rather than for a mutation in an specific individual <- rather than for a mutation in a specific individual

We have resolved this grammatical error - “However, this method only predicts clonal expansion rate for a mutation aggregated over a population rather than for a mutation in a specific individual.”

Page 4: CH is common and increases with age <- CH is not common in young individuals so this reads weird

The text now reads “CH is common in the elderly and prevalence increases with age.”

Page 4: Previous studies have shown that CH significantly increases an individual’s risk of all-cause mortality and many common complex diseases <- many of these studies show associations, not causations, so it is incorrect that CH increases the risk

The text now reads “Previous studies have shown that CH is associated with increased risk of all-cause mortality and many common complex diseases.”

Page 4: mCAs ... occur in about 3% of individuals older than 50 years old without cancer <- the rate of detectable mCAs is much higher after age 50, even if you only restrict to autosomal mCAs (see Terao, C. et al . Chromosomal alterations among age-related haematopoietic clones in Japan. Nature (2020))

The text now reads “occur in between 10-20% of individuals over 55 years old without cancer (Terao et al, Nature, 2020)(Loh et al, Nature, 2020).”

Page 4: gain or loss of an entire chromosome <- when you restrict to the autosomes, loss of an entire chromosome in blood is almost never observed (likely it is not tolerated) so this sentence could be rewritten

The text now reads “mCAs, which may involve a gain or loss of a >1Mb segment of a chromosome or a copy-neutral loss of heterozygosity (CN-LOH),...”

Page 4: Although individuals with larger mCA clones generally have worse health outcomes <- I think you mean larger cell fraction clones, but this reads as mCAs that are larger so rewrite this. Also, I think not all the references in this sentence support this claim so make sure you reference supports worse health outcomes for larger cell fractions

The text now reads “Although individuals with CH clones of higher cell fraction (i.e., larger clone size) generally have worse health outcomes...”

Additionally, we verified the citations and have included the text below from each citation that supports worse health outcomes for larger cell fractions.

Loh et al: “Individuals with incident CLL exhibited clonality up to six years before diagnosis, and clonal fraction inversely correlated with time to malignancy (Fig. 5c).”

Jaiswal et al (2014): “Among persons with a variant allele fraction of 0.10 or greater (indicating a higher proportion of cells in the blood carrying the mutation), the risk of a hematologic cancer was increased by a factor of nearly 50 (hazard ratio, 49; 95% CI, 21 to 120; $P < 0.001$).”

Jaiswal et al (2017): “Therefore, we tested whether CHIP with a larger clone size was also associated with a greater burden of atherosclerosis. CHIP carriers without incident coronary heart disease but with a variant allele fraction of a least 10% had 12 times the risk of having a coronary-artery calcification score of 615 or more as did noncarriers ($P = 0.002$ by logistic regression after adjustment), whereas participants with a variant allele fraction of less than 10% had no increased risk ($P = 0.02$ for heterogeneity).”

Zekavat et al: “The dependence of this association with mCA cell fraction is further visualized in Figure 3a,b which shows an increase in proportion of incident infection cases and incident sepsis cases with cell fraction, with greater slopes observed at approximately cell fraction $> 10\%$.”

“Similarly, incident hematologic cancer risk was also strongly dependent on cell fraction (Figure 2c). We reproduced the associations of mCAs with hematologic cancers with similar effects as previously described in the UK Biobank. We found that expanded autosomal mCAs with cell fraction $> 10\%$ were most strongly associated with incident hematologic cancer (Figure 2d).”

Page 4: so obtaining sufficient sample sizes is challenging <- this is only challenging because we are using datasets generated for reasons other than testing for CH so this needs to be better explained

The text now reads “*Conventional methods to study clonal fitness require collecting serial blood samples over several decades, so obtaining sufficient sample sizes from existing biobanks is challenging as they typically only include one blood sample per individual.*”

Page 4: Estimating the fitness of a given driver mutation <- even in the original PACER paper the fitness was computed for mutated driver genes rather than single mutations. Although that might be the goal, I find it confusing as the fitness is always computed about a class of mutations

PACER scores are computed for an individual; these PACER scores can be aggregated (e.g., median) for all individuals with a given class of mutations to represent the fitness of a class of mutations.

We have updated the sentence to state: “*Estimating the clonal expansion rate of a given driver mutation in an individual is essential to elucidate germline modifiers of clonal expansion and better characterize the pathophysiology of CH-associated diseases.*”

Page 4: to estimate the fitness of a SNV with a single blood sample <- this is incorrect as in the original PACER paper you do not estimate the fitness of a SNV but classed of SNV (binned by driver gene)

Thanks for this comment. Passenger mutations are counted per-individual and then adjusted for an individuals' covariates (e.g., age, sex, study, etc.). Therefore, PACER score estimates clonal expansion rate of an SNV in an individual. When these values are aggregated across individuals with a class of SNV (or type of mCA), they estimate the fitness of a class of mutations based on PACER scores of observed individuals with that class of mutation.

The referenced sentence now states: *“To overcome this limitation, methods have been developed to estimate the expansion rate of a clone in an individual from a single blood draw.”*

Page 4: we apply PACER to 6,381 individuals with gain, loss, and CN-LOH mCAs in the NHLBI Trans-Omics for Precision Medicine (TOPMed) dataset <- given the right assay, anybody could be classified as having gain, loss, and CN-LOH so it is important to specify that these are events detected with a given assay (WGS)

The text now reads *“Here, we apply PACER to 6,381 individuals with gain, loss, and CN-LOH mCAs, detected by whole genome sequencing (WGS), in the NHLBI Trans-Omics for Precision Medicine (TOPMed) dataset to identify determinants and consequences of mCA fitness (Fig 1a).”*

Page 4: using our individual mCA fitness estimates <- the fitness is for mCA classes ... I think what you mean here is PACER scores

Thank you for pointing this out. You are absolutely correct and we have adjusted the text to now say this. The paragraph has been rewritten significantly and now reads:

“We recently developed a method called passenger-approximated clonal expansion rate (PACER), which uses the abundance of passenger mutations accompanying a driver mutation to estimate the clonal expansion rate of a SNV with a single blood sample from an individual.²⁰ Here, we apply PACER to 6,381 individuals with gain, loss, and CN-LOH mCAs, detected by whole genome sequencing (WGS), in the NHLBI Trans-Omics for Precision Medicine (TOPMed) dataset to calculate a PACER score per individual and identify determinants and consequences of mCA clonal expansion rate (Fig 1a). PACER estimates of mCA fitness, derived by calculating the median of PACER scores across all individuals with a specific mCA, were compared to the fitness of SNV mutations implicated in CHIP (Extended Data Fig 1). We examined associations between PACER score and peripheral blood counts and observed that for individuals with single nucleotide variants or mCAs affecting JAK2, higher PACER score (i.e., faster mCA expansion rate) associated with higher erythrocyte counts. Next, we performed a genome-wide association study (GWAS) of PACER score among individuals with a single mCA and found that variants in TCL1A, NRIP1, and TERT may modulate mCA clonal expansion.”

Page 5: We identified 6,930 people with one mCA and 763 with multiple mCAs <- how many had no mCAs detected?

As referenced in Jakubek et al 2023's publication of mCA calls in TOPMed, 67,390 participants with WGS data were assessed for mCAs with the MoChA v1.11 caller. Out of these participants, 6,930 people had one mCA and 763 had multiple mCAs. Therefore, 59,697 (88%) had no mCAs detected.

We have adjusted the methods to say this: "Using WGS data from 67,390 individuals in TOPMed, we identified 7,693 individuals with mosaic chromosomal alterations using MoChA. Of those individuals, 6,930 people had a single mCA and 763 with multiple mCAs."

Reference: Jakubek, Y. A. et al. Mosaic chromosomal alterations in blood across ancestries using whole-genome sequencing. Nat. Genet. 55, 1912–1919 (2023).

Page 5: After excluding mCAs of sex chromosomes, we detected 3,828 autosomal mCAs in 3,068 unique individuals <- how many of these individuals had detected sex chromosome mCAs?

Thank you for this question. 571 individuals with detected autosomal mCAs also had detectable sex chromosomal mCAs (i.e., loss of chromosome X or Y). 369 were mLOX, and 202 were mLOY.

The text has been updated to reflect this information: "571 individuals with detected autosomal mCAs also had detectable sex chromosomal mCAs (i.e., loss of chromosome X or Y)."

Page 5: the minimum total passenger mutations ... was <- the minimum total passenger mutations count ... was

The text now reads "Within this cohort, the minimum total passenger mutation count for a patient with an mCA was 3, maximum was 933, and median was 53 (Fig 2a)."

Page 5: We calculated a covariate-adjusted PACER <- We calculated a covariate-adjusted PACER score

The text now reads "We then calculated a covariate-adjusted PACER score to approximate clonal expansion rate for the mCA of each individual from the normalized residuals of a negative binomial regression of age, sex, and clonal fraction predicting total passenger mutation count."

Page 5: we compared passenger mutation counts between mCA chromosomes <- I understand what you mean here but you need to define first what you mean with the term mCA chromosomes

The text now reads "To test this, we compared passenger mutation counts between the chromosome containing the mCA to the chromosomes not containing the mCA and found no significant difference by mCA location and type (Extended Data Fig 1a) or in aggregate ($p = 0.12$)."

Page 5: The most common autosomal mCAs were on chromosomes 11, 1, and 9 <- I bet these are mostly 1q CN-LOHs driven by MPL, 9p CN-LOHs driven by JAK2, and 11q CN-LOHs driven by ATM. However, as the data is reported, the less experienced reader will have no way to make this determination. It would be helpful here to be a bit more specific

We have added to the discussion in order to provide more context on this finding.

The text now reads “We observed high prevalence of mCAs on chromosomes 1, 9, and 11 – specifically CN-LOH. The likely explanation for increased prevalence of these mCAs is that these chromosomes contain the genes MPL, JAK2, and ATM respectively and somatic mutations in these genes make these mCAs more detectable due to their increased proliferative ability.”

Page 5: with CN-LOH being the majority (52%). Clonal fraction showed significant variation across autosomal mCA types and chromosomes (Fig 1c) <- A lot of reader will not understand that this is at least partially explained by CN-LOHs being much easier to detect both because of their larger size and because they affect two haplotypes at once. Similarly gains are easier to detect than losses as they tend to be larger. It must be made clear that cell fraction differences are mostly driven by limits of detection rather than by differences in biological processes. It's also not clear how by looking at Fig 1c you can assess that there is significant variation in clonal fractions

Thank you for these insights on the detectability of mCAs by type and the interpretability of the variations in clonal fraction. We appreciate this comment, as we did not precisely state our results regarding Fig 1c – instead we now write “mCAs within the same type and chromosome exhibited high variability in clonal fraction” – our interpretation is focused on the variability in clonal fraction within the same type of mCA rather than comparing mCA types, as we agree there is not a clear way to compare variation in clonal fractions across mCAs in Fig 1c.

Moreover, we added a paragraph to the discussion describing the varying prevalence of mCAs by mutation type and chromosome:

“The most commonly observed mCAs involved chromosomes 1, 9, and 11 – specifically CN-LOH in these chromosomes. The likely explanation for increased prevalence of these mCAs is that these chromosomes contain the genes MPL, JAK2, and ATM respectively and somatic mutations in these genes make these mCAs more detectable due to their increased proliferative ability. Across mCA types in TOPMed, CN-LOH mutations were the most common, likely due to the increased detection ability to detect CN-LOH events due to their larger size and their effect on both haplotypes.”

Page 5: we quantified mCA fitness in individuals with a single mCA and no CH-associated SNVs <- my understanding is that fitness is a statistic assigned to a class of mCAs, not to an individual ... there is quite some confusion in the writing

The text has been updated and now reads “We calculated passenger mutation counts, representing clock-like C>T or T>C somatic mutations (see Methods), in individuals with a single mCA and no CH-associated SNVs.”

Page 5: the minimum total passenger mutations for a patient with an mCA was <- at this point passenger mutations are not defined. Although they are then later defined in the methods, there is no "see Methods" notice anywhere before explaining that there is a definition later on. As I was reading the paper I had no idea what a passenger mutation was, especially given the very specific definition

We now reference the methods section in this sentence, which contains more detail about passenger mutations and what they represent. The text reads: “We calculated passenger mutation counts, representing clock-like C>T or T>C somatic mutations (see Methods), in individuals with a single mCA and no CH-associated SNVs.”

Page 5: We calculated a covariate-adjusted PACER <- before this instance PACER is described as a method while here it seems to be a statistic or a score ... this is confusing. Again, make sure everything is properly defined before being used. Do you mean here a statistic being equal to the PACER fitness of the mCA carried by the individual after adjusting for covariates?

We have clarified that we are calculating a per-individual PACER score which is a covariate-adjusted estimate of clonal expansion rate. The text now reads “We then calculated a covariate-adjusted PACER score to approximate clonal expansion rate for the mCA of each individual from the normalized residuals of a negative binomial regression of age, sex, and clonal fraction predicting total passenger mutation count.”

Page 5: As a sensitivity analysis <- Maybe you meant something like "to check for potential confounders"

The text now reads “To check for potential confounders, we investigated if these larger rearrangements alter total passenger mutation counts, thereby confounding our estimations of clonal expansion rate.”

Page 5: Furthermore, we found that PACER scores <- notice here you use PACER score but this is never defined

We have now defined a PACER score in the previous paragraph with the following text “We then calculated a covariate-adjusted PACER score to approximate clonal expansion rate for the mCA of each individual from the normalized residuals of a negative binomial regression of age, sex, and clonal fraction predicting total passenger mutation count.”

Page 5: than driver SNVs in non-R882 DNMT3A <- why choose this one from the PACER paper? Maybe explain the rationale behind this

Non-R882 DNMT3A was chosen as the reference because these are typically the slowest growing CHIP mutations and therefore serve as a baseline standard for comparison. We have clarified this in the text “Of all mCAs, only a gain in chromosome 1 had a higher fitness than driver SNVs in non-R882 DNMT3A, which is the slowest growing CHIP mutation. (Weinstock et al., Nature, 2023)”

Page 5: A generalized linear model of median PACER <- is this the median of the PACER scores across individuals carrying an mCA of the same mCA class?

Yes, precisely. We computed the median of PACER scores for individuals with the same mCA type and location to derive a measure of an mCA's fitness from PACER. This procedure was performed because the clonal-fraction-derived fitness from Watson and Blundell, 2023 derives a fitness estimate for each

mCA chromosome and type rather than per individual; therefore, we aggregated our per-individual estimates to derive a value per mCA chromosome and type for the comparison.

We added a section in the methods called “Aggregation of per-individual PACER scores to calculate PACER-derived mCA fitness” and included the following sentence: “Multiple linear regression was performed between clonal-fraction-derived fitness from Watson and Blundell, 2023 and PACER-estimated mCA fitness, with median age of individuals with the mCA chromosome and type as a covariate.”

Page 6: , individuals with lymphoid mCAs within TOPMed had a significantly ($p = 0.0004$) higher lymphocyte count ... than those with non-lymphoid mCAs <- report here the number of individuals with lymphoid mCAs and those with non-lymphoid mCAs. Also, how did you count the individuals that had both?

Thanks for this clarifying question. Of individuals with autosomal mCAs, 98 individuals had lymphoid mCAs, and 889 did not. Since we only included individuals with a single mCA for our PACER method, there were no individuals with both lymphoid and nonlymphoid mCAs.

However, we have changed this analysis from the paper, as our correlation between PACER score and lymphocyte counts in the multiple regression was driven by outliers (as per your next comment) with lymphocyte counts $> 10 \times 10^9$ cells/L. We did apply your suggestion more generally and provided for each regression the number of individuals in each category of mCA with blood counts available to provide this necessary information to the reader to interpret each regression, as we show below:

“For the 23 individuals with lymphoid mCAs who had lymphocyte counts...”

“For the 47 individuals with myeloid mCAs who had myeloid cell counts...”

“for 11 individuals with CN-LOH or loss of chromosome 9p and erythrocyte counts.”

The full text is pasted in the next comment with our complete analysis.

Page 6: Among individuals with lymphoid mCAs, a multiple regression of age at time of blood draw, sex, clonal fraction, and PACER predicting lymphocyte count explained 13.3% of the variance in lymphocyte counts, compared to only 2.5% for a model without PACER (Fig 3a) <- by the figure it seems as if this might be driven by a few outliers ... what is the p-value associated with this increase? (I see this is reported later as $p=0.019$... maybe just incorporate the two sentences so that it is clear that the statement is only nominally significant)

We thank the reviewer for bringing up this excellent point. As you point out, when we exclude individuals with lymphocyte counts greater than 10, the PACER score is no longer significantly associated with lymphocyte count among individuals with lymphoid mCAs. Given the limited number of individuals with complete blood counts available in TOPMed, it is likely we are underpowered to detect a correlation between PACER score and blood counts. However, given the strong correlation between PACER scores and erythrocyte counts among individuals with copy-neutral loss of heterozygosity or loss of the p arm of chromosome 9 (9p- and 9p- respectively), we further investigated associations between

PACER score and erythrocytes to validate and understand the clinical significance of the approximated expansion rate from passenger mutation count.

To do this, we compared the association between PACER score and erythrocyte counts among individuals with 9p= and 9p- mCAs to the same association among individuals with JAK2 V617F clonal hematopoiesis of indeterminate potential (CHIP). To do this, we performed multiple linear regression of age at time of blood draw, sex, clonal fraction, and PACER to predict erythrocyte count among individuals with JAK2 V617F CHIP in the same manner done for mCAs associated with polycythemia vera (9p= and 9p-). We observed a significant association ($\beta = 0.341$, 95% CI [0.133, 0.537], $p = 0.001$), and the model with PACER score included had an R^2 of 0.90 while a model with only age and sex had an R^2 of 0.32. We now have included this plot in Figure 3. These associations include individuals with either a detected mCA or detected JAK2 V617F CHIP, as we only apply PACER scores to individuals with a single clonal driver.

We believe this plot demonstrates how PACER scores reflect the phenotypic consequences of mCA and CHIP clonal expansion rate for erythrocyte counts. We therefore have updated Figure 3 to show a scatterplot of erythrocyte counts (10¹² cells/L) versus covariate-adjusted passenger associated clonal expansion rate (PACER) score for (a) patients with copy-neutral loss of heterozygosity or loss of the p arm of chromosome 9 and (b) patients with JAK2 V617F clonal hematopoiesis of indeterminate potential.

Accordingly, we added the following to our methods to describe where we derived these JAK2 V617F calls: *“The calls of somatic singletons for clonal hematopoiesis were made from the publicly available data from Bick et al, 2020. Individuals with JAK2 V617F CHIP mutations were also determined with this dataset.”*

Therefore, we have deemphasized the finding of the correlation of PACER score and lymphocyte count among individuals with lymphoid-cancer-associated mCAs and instead replaced it with the correlation of

PACER score and erythrocyte count among individuals with mCAs affecting chromosome 9 and with JAK2 V617F CHIP.

The new results section now reads:

“For the 23 individuals with lymphoid mCAs who had lymphocyte counts, a multiple regression of age at time of blood draw, sex, clonal fraction, and PACER score predicting lymphocyte count, demonstrated a significant association between PACER score and lymphocyte counts ($\beta = 0.0175$, 95% CI [0.003, 0.032], $p = 0.019$). However, after excluding outliers with lymphocyte count greater than 10×10^9 cells/L, there was no significant association. These outliers may represent elevated lymphocyte counts in three individuals with a high PACER score due to rapidly expanding mCA clones.

For the 47 individuals with myeloid mCAs who had myeloid cell counts, a multiple regression of age at time of blood draw, sex, clonal fraction, and PACER score predicting myeloid cell count found that PACER score was not significantly associated with myeloid cell count ($\beta = -0.0031$, 95% CI [-0.009, 0.003], $p = 0.298$).

Since mutations in JAK2, such as JAK2 V617F, on chromosome 9p are known to cause polycythemia vera, we performed a multiple regression of age at time of blood draw, sex, and PACER score to predict erythrocyte counts for 11 individuals with CN-LOH or loss of chromosome 9p and erythrocyte count. This model explained 91.6% of the variance in erythrocyte count, and PACER score had a significant positive association with erythrocyte count ($\beta = 0.0119$, 95% CI [0.006, 0.018], $p = 0.018$) (Fig 3a).

We then examined the association between PACER score and erythrocyte counts among individuals with JAK2 V617F clonal hematopoiesis of indeterminate potential (CHIP). We performed multiple linear regression of age at time of blood draw, sex, clonal fraction, and PACER to predict erythrocyte count among individuals with JAK2 V617F CHIP and observed a significant association ($\beta = 0.341$, 95% CI [0.133, 0.537], $p = 0.001$), and the model with PACER score included had an R^2 of 0.90 while a model with only age and sex had an R^2 of 0.32. Therefore, for individuals with mCAs or somatic SNVs affecting JAK2, erythrocyte count is associated strongly with PACER score.”

Page 10: MoChA is a method that identifies mCAs using three hidden Markov models to find mCA-induced deviations in allelic balance at heterozygous sites <- this is incorrect as stated at <https://github.com/freeseek/mocha#hmm-parameters> where it says "There are a few differences between MoChA and the HMM model used in Loh et al. 2018, Thompson et al. 2019, Loh et al. 2020, and Terao et al. 2020. The most important is that, the latter model used a likelihood ratio test statistic with likelihoods deriving from a 3-state forward-backward HMM model. MoChA instead uses a Viterbi HMM model with multiple alternate states and it increases the number of alternate states dynamically when trying to assess multiple calls". MoChA tests a large array of different cell fractions as multiple hidden states all at once. Although essentially similar to what done in the original studies, it does not run a three states HMM

Thank you for your comments regarding our description of MoChA. We have updated our methods on mCA detection significantly and have removed the reference to a 3-state HMM.

“Using WGS data from 67,390 individuals in TOPMed, we identified 7,693 individuals with mosaic chromosomal alterations using MoChA version 1.11.21 MoChA relies on haplotype-phasing to detect mCAs; we performed haplotype phasing with SHAPEIT4. Using these phased genotypes, MoChA evaluates coverage and B allele frequency (BAF) at heterozygous loci to detect mCAs. Heterozygous markers from Taliun et al were used. The MoChA tool was executed with the additional parameter ‘–LRR-weight 0.0–bdev-LRR-BAF 6.0’, which deactivated the LRR + BAF model. MoChA is a method that identifies mCAs to find mCA-induced deviations in allelic balance at heterozygous sites, as described previously.^{5,13} An mCA was defined as a gain, loss, or copy-neutral loss of heterozygosity in a specific chromosome and p or q arm. Code is available at <https://github.com/freeseek/mocha>.”

We excluded 160 samples with phased BAF auto-correlation >0.05, indicative of contamination or other potential sources of poor DNA quality, and 67 samples with phenotype-genotype sex discordance. We removed likely germline copy number polymorphisms (lod_baf_phase <20 for autosomal variants and lod_baf_phase <5 for sex chromosome variants), constitutional or inborn duplications (mCAs 2-10 Mb with relative coverage >2.25, and mCAs 50-250 Mb with relative coverage >2.5) and deletions (filtering out mCAs with relative coverage <0.5). We defined a threshold of minimum mCA size at 2 Mb and excluded mCAs with size of < 2 Mb. We defined mosaic loss of the X chromosome (X-) as a loss of a segment of chromosome X > 100 Mb in size and with a relative coverage < 2.5. Of those individuals, 6,930 people had a single mCA and 763 with multiple mCAs.”

Page 10: As was done in Weinstock et al, passenger mutations were defined as C>T and T>C base-pair substitutions that do not occur in a CH-associated gene. These mutations are “clock-like” and are associated with aging, whereas other mutations may be affected by disease or environmental factors <- is this because most C>T and T>C base-pair substitutions are CpG mutations? If so, was there a reason why you didn't simply look at CpG mutations (i.e. transitions at CpG sites)?

C>T and T<C mutations were most strongly age-associated in the cohort (Weinstock et al., *Nature*, 2023) and displayed “clock-like” signatures in HSCs (Alexandrov et al., *Nature Genetics*, 2015). We have updated the methods section to provide this clarification.

The text now reads “*Since HSCs acquire neutral passenger mutations, defined as age-associated “clock-like” C>T and T>C base-pair substitutions, (Alexandrov et al., Nature Genetics, 2015) at a fairly consistent rate across individuals, (Osorio et al., Cell Reports, 2018; Mitchell et al., Nature, 2022; Williams et al., Nature, 2022) these mutations can be used as a proxy for the passage of time to approximate when a CH driver mutation was acquired.*”

Page 11: Linkage disequilibrium and conditional analysis for rs1122138 <- explain in this section that these are two variants in the TCL1A gene

The text now reads “*To determine whether in TCL1A rs1122138 is a distinct signal from rs2887399, a previously reported SNP associated with clonal expansion of SNV CH.*”.

Page 13: GWAS summary statistics for PACER amongst individuals with mCAs are available in the following GitHub repository here: <https://github.com/bicklab/pacer-mca->

fitness/blob/main/Data/gwas_results_pacerint_nochip_df.assoc <- I assume this will become available once the paper is published as at the moment there is no pacer-mca-fitness repository

Thank you for pointing this out - we had forgotten to switch the repository to public. Our Github repository is now public.

Page 15: Scatter plot of median total passenger mutations and fitness derived from clonal fraction by mCA <- this is a bit confusing as the x-axis is CF-derived fitness, not median total passenger mutations ... I am confused about what is actually being plotted here

We have clarified how these metrics were derived in the Methods section, as described above – specifically how PACER scores are aggregated by mCA chromosome and type to enable comparison with clonal-fraction-derived fitness for an mCA by Watson and Blundell in the UK Biobank. This graph is comparing CF-derived fitness with PACER-estimated fitness and shows a high degree of correlation between them.

We have also updated the figure caption to be more clear:

“c) Scatter plot of median total passenger mutations and fitness derived from clonal fraction by mCA by Watson and Blundell 2023 (CF-derived fitness) for mCAs with >25 individuals with a given mCA type. The size of the dot corresponds to the number of individuals with that mCA type. A generalized linear model of PACER score, mean age, and mean clonal fraction predicting CF-derived fitness had an R^2 value of 0.49.”

Page 15: only includes mCAs with counts > 25 <- what is counts? I assume this is the number of individuals with a given mCA type, so say so and don't interchangeably use mCA and mCA type as it is hard to understand what you are plotting here ... mCA is one observed event, not a class of events

The text has been updated to read *“Scatter plot of median total passenger mutations and fitness derived from clonal fraction by mCA (only includes mCAs with >25 individuals with a given mCA type).”*

Page 23: Zekavat, S. M. et al. Hematopoietic mosaic chromosomal alterations and risk for infection among 767,891 individuals without blood cancer. medRxiv 2020.11.12.20230821 (2020)
doi:10.1101/2020.11.12.20230821.599 <- this preprint has now been published in Nature M

The citation has been updated to reflect the published version.

Page 23: Watson, C. J. & Blundell, J. R. Mutation rates and fitness consequences of mosaic chromosomal alterations in blood. 2022.05.07.491016 Preprint at <https://doi.org/10.1101/2022.05.07.491016> (2022) <- this preprint has now been published in Nature Genetics

The citation has been updated to reflect the published version.

Figure 1b: please consider a pileup plot or a circos plot or anything that can readily distinguish mCAs by size and by chromosome arm as a histogram leaves a lot of information out (Extended Data Fig. 1a,b of Zekavat et al. 2021 is an example)

Thank you for this suggestion. We completely agree that visualizing the plot in this way is helpful to see the distribution of mCAs across the chromosomes. Since we use the same mCA calls as Jakubek et al, 2023, we have updated the manuscript to reference this figure specifically in the results section. We have added the sentence: “Visualization of the mCAs detected in TOPMed are shown in Figure 1 of Jakubek et al, 2023.”

Figure 2b: the Y-axis is labeled as "fold change" but should it really be log fold-change? And what log? Natural, 2, or 10?

Thank you for allowing us to clarify what we mean by fold-change. We computed the median of PACER scores for individuals with the same mCA type and location to derive a measure of an mCA's fitness from PACER. With this estimation of mCA fitness derived from PACER scores, we then calculated the fold-change relative to a loss of chromosome X, defined as a loss of a > 100 Mb segment of chromosome X.

We have changed the methods to describe this process:

“To derive the PACER-estimated mCA fitness for a given mCA chromosome and type, we computed the median of PACER scores for all individuals with a given mCA type and chromosome. We then calculated the fold change for each estimate of mCA fitness relative to loss of the X chromosome (loss of > 100 Mb segment of chromosome X) by taking the ratio of the mCA fitness and the fitness of loss of the X chromosome.”

Additionally, the Fig 2b caption now has this clarification as well:

“The PACER scores are calculated after covariate adjustment (age, sex, study cohort, and clonal fraction) and inverse normalization of the total passenger mutations for all individuals with a single mCA. The median of the PACER scores is computed for individuals with the same mCA type and location to estimate mCA fitness. The fold change in estimated mCA fitness is calculated by dividing the clonal expansion rate for a given mCA by that for a loss of chromosome X.”

Extended Data Fig 2: Covariate-adjusted PACER score <- notice here you use PACER score but this is never defined

Thank you for this comment. We have updated the manuscript to define this quantity more clearly.

In the results section: “We then calculated a covariate-adjusted PACER score to approximate clonal expansion rate for the mCA of each individual from the normalized residuals of a negative binomial regression of age, sex, and clonal fraction predicting total passenger mutation count.”

In the methods section: “After the number of total passenger mutations was calculated, we fit a negative binomial regression model of age, sex, and clonal fraction to predict total passenger mutations using scikit-learn in Python. We then performed a Yeo-Johnson inverse-normal transformation on the residuals

using the SciPy package in Python 2.7.17. We then used these covariate-adjusted residuals to calculate PACER score.”

REVIEWERS' COMMENTS

Reviewer #1 (Remarks to the Author):

The authors have addressed all of my comments and suggestions satisfactorily.

Reviewer #2 (Remarks to the Author):

The revised version of the manuscript reads much better. The additional explanation in the PACER section in the methods was particularly helpful to clarify my confusion. I very much enjoyed reading this study

Minor comments:

Page 5: found that variants in *TCL1A*, *NRIP1*, and *TERT* may modulate mCA clonal expansion <- gene names are not italicized

Page 5: Of detectable mCAs, CN-LOH was the majority <- Of detectable mCAs, CN-LOH mCAs were the majority

Page 6: to those generated by a another approach <- to those generated by another approach

Page 9: likely due to the increased detection ability to detect <- likely due to the increased ability to detect

Page 9: the pro-proliferative *TCL1A* gene <- gene name is not italicized

Page 9: somatic rearrangements in *TCL1A* <- gene name is not italicized

Page 13: Since the detection limit of WGS at x38 coverage <- Since the detection limit of WGS at 38x coverage(?)

Reviewer #1 (Remarks to the Author):

The authors have addressed all of my comments and suggestions satisfactorily.

Response: We thank the referee for their excellent suggestions and comments in the prior review stage, as we feel they have greatly improved the quality and reproducibility of our paper.

Reviewer #2 (Remarks to the Author):

The revised version of the manuscript reads much better. The additional explanation in the PACER section in the methods was particularly helpful to clarify my confusion. I very much enjoyed reading this study

Minor comments:

Page 5: found that variants in *TCL1A*, *NRIP1*, and *TERT* may modulate mCA clonal expansion <- gene names are not italicized

Page 5: Of detectable mCAs, CN-LOH was the majority <- Of detectable mCAs, CN-LOH mCAs were the majority

Page 6: to those generated by a another approach <- to those generated by another approach

Page 9: likely due to the increased detection ability to detect <- likely due to the increased ability to detect

Page 9: the pro-proliferative *TCL1A* gene <- gene name is not italicized

Page 9: somatic rearrangements in *TCL1A* <- gene name is not italicized

Page 13: Since the detection limit of WGS at x38 coverage <- Since the detection limit of WGS at 38x coverage(?)

Response: We greatly appreciate the referee for their meticulous suggestions and comments throughout the review process. We are happy to hear that the manuscript reads better after our changes. Each of these exact grammatical changes have now been made in the latest version of the manuscript.